# DISCOUNT MODEL SEARCH FOR QUALITY DIVERSITY OPTIMIZATION IN HIGH-DIMENSIONAL MEASURE SPACES

**Bryon Tjanaka,[1] Henry Chen,[1] Matthew C. Fontaine,[1] Stefanos Nikolaidis[1,2]**
[1]Thomas Lord Department of Computer Science, University of Southern California
[2]Archimedes AI
{tjanaka,hchen365,mfontain,nikolaid}@usc.edu

## ABSTRACT

Quality diversity (QD) optimization searches for a collection of solutions that optimize an objective while attaining diverse outputs of a user-specified, vector-valued measure function. Contemporary QD algorithms are typically limited to low-dimensional measures because high-dimensional measures are prone to distortion, where many solutions found by the QD algorithm map to similar measures. For example, the state-of-the-art CMA-MAE algorithm guides measure space exploration with a histogram in measure space that records so-called discount values. However, CMA-MAE stagnates in domains with high-dimensional measure spaces because solutions with similar measures fall into the same histogram cell and hence receive the same discount value. To address these limitations, we propose Discount Model Search (DMS), which guides exploration with a model that provides a smooth, continuous representation of discount values. In high-dimensional measure spaces, this model enables DMS to distinguish between solutions with similar measures and thus continue exploration. We show that DMS facilitates new capabilities for QD algorithms by introducing two new domains where the measure space is the high-dimensional space of images, which enables users to specify their desired measures by providing a dataset of images rather than hand-designing the measure function. Results in these domains and on high-dimensional benchmarks show that DMS outperforms CMA-MAE and other existing black-box QD algorithms.

## 1 INTRODUCTION

We present a method that enhances exploration in quality diversity (QD) optimization and show how this method enables new applications for QD.[1] QD (Pugh et al., 2016) is a branch of stochastic optimization that seeks to find diverse, high-performing solutions to a problem, with applications like robotics (Mouret & Clune, 2015), generative modeling (Ding et al., 2024), and LLM red-teaming (Samvelyan et al., 2025). To illustrate, consider searching for "a photo of a hiker." By itself, this problem is ambiguous since hikers vary widely in appearance, depending on, for instance, where they are hiking and the time of year. If we run a single-objective optimization algorithm like Adam (Kingma & Ba, 2015) or CMA-ES (Hansen, 2016), the image we find would optimize the objective $f$ of "a photo of a hiker," but the hiker could take on one of many different appearances. In contrast, QD (Pugh et al., 2016) can manage the ambiguity by searching for an archive (set) of images that both optimize the objective $f$ and diversify along the outputs of a measure function $m$.

Prior work (Pugh et al., 2016) typically hand-designs $m$ to output low-dimensional ($<10$D) vectors. To illustrate, our measure function could output two measures: the hiker's age and the temperature for which they are dressed. Then, our archive would contain images like a younger hiker dressed for cold weather and an older hiker dressed for warm weather, as well as all images in between.

One reason prior works focus on low-dimensional measures is that high-dimensional measure spaces are prone to *distortion*, where many solutions (images) map to a small region of measure space

---

[1]Project page available at https://discount-models.github.io

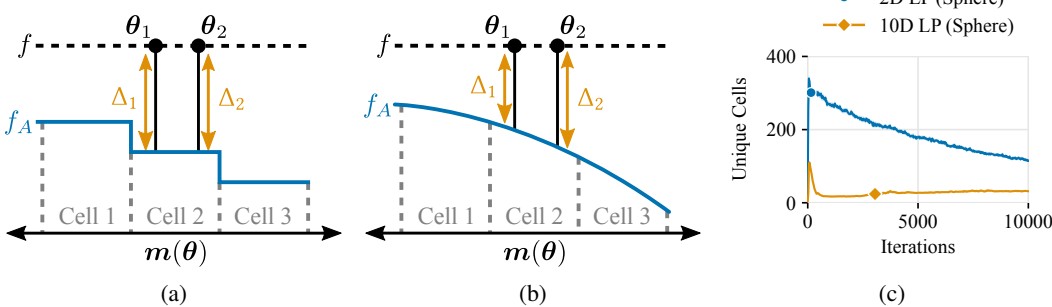

Figure 1: (a): One failure mode of CMA-MAE. On a flat objective $f$, solutions $\theta_1$ and $\theta_2$ fall in the same archive cell based on their measures, resulting in identical discount values from the discount function $f_A$. (b): In our proposed DMS, the discount model provides a smooth discount function that assigns distinct discount values to $\theta_1$ and $\theta_2$, showing that $\theta_2$ has greater archive improvement than $\theta_1$ ($\Delta_2 > \Delta_1$) and thus providing a stronger signal to guide search. (c): Number of unique cells where solutions sampled by CMA-MAE land in two benchmarks (mean over 20 trials; Sec. 4).

(i.e., the solutions have similar measures). For example, it may be easy to search for images of hikers in warm weather, while hikers in cold weather are hard to find. Although distortion exists in low-dimensional measure spaces (Fontaine & Nikolaidis, 2021; Fontaine et al., 2020), it is more prominent in high-dimensional measure spaces because there are exponentially larger volumes to which solutions can map (Sec. 4).

We propose to scale to high-dimensional measure spaces by addressing the effects of distortion in Covariance Matrix Adaptation MAP-Annealing (CMA-MAE) (Fontaine & Nikolaidis, 2023), a state-of-the-art black-box QD algorithm. To fill the archive of solutions, CMA-MAE searches for solutions $\theta$ that maximize *archive improvement*, defined as $\Delta(\theta) = f(\theta) - f_A(m(\theta))$, where $f(\theta)$ is the solution's objective value and $f_A$ is a *discount function* that returns scalar *discount values* based on the solution's measures $m(\theta)$. CMA-MAE represents $f_A$ as a histogram by tessellating the measure space into cells and storing a scalar value in each cell. In domains[2] with high distortion, particularly domains with high-dimensional measures, solutions with similar measures fall into the same cell. As a result, CMA-MAE incorrectly assigns these solutions the same discount value, which creates inaccurate improvement values that cause the search to stagnate (Fig. 1a).

*Our key insight is that a smooth, continuous representation of the discount function will enhance exploration in high-dimensional measure spaces.* As such, we propose Discount Model Search (DMS), where a smooth *discount model* accurately assigns distinct discount values to solutions even when distortion causes them to have similar measures (Fig. 1b). The discount values guide DMS to explore the measure space and discover solutions long after CMA-MAE would stagnate.

We show that by scaling to high-dimensional measure spaces, DMS facilitates new capabilities for QD. For example, it can be challenging to design a low-dimensional measure function to describe "where the hiker is located," as locations vary widely from beaches to mountains to forests. In general, creating measure functions can be tedious and unintuitive — similar to objective functions (Skalse et al., 2022; Lehman et al., 2020), the design of the measure function vastly affects the quality of solutions produced (Bossens et al., 2020; Pugh et al., 2015). In contrast, consider that by defining the measures as age and temperature in our earlier example, we essentially specified a 2D grid of (age, temperature) points for which we sought images of hikers. If we treat the high-dimensional space of images (e.g., $256 \times 256 \times 3$ RGB vectors) as the measure space, we can replace the 2D points with images, i.e., we can easily specify "where the hiker is located" by providing a dataset of landscape images (Fig. 2). Thus, by scaling to high-dimensional measure spaces, we believe DMS makes QD more accessible: it enables specifying measures via datasets, which can alleviate the need for manual measure function design and enable specifying new types of measures. We refer to this setup as *Quality Diversity with Datasets of Measures (QDDM)*.

---

[2]In our work, the term "domain" is synonymous with "problem."

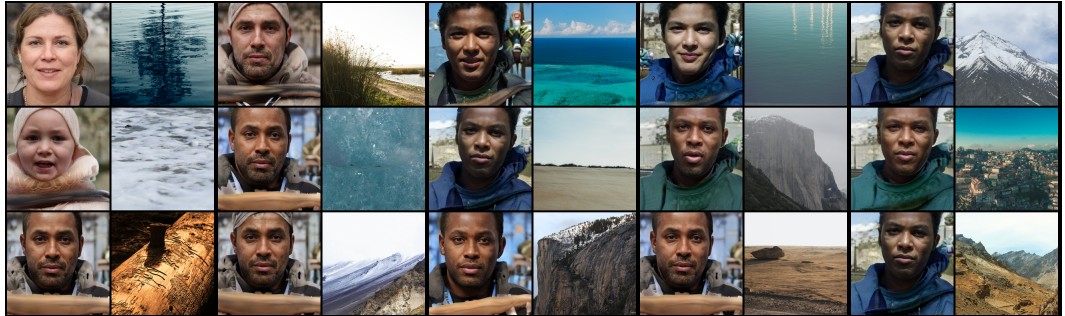

Figure 2: In the LSI (Hiker) domain, the objective is "A photo of the face of a hiker," and the measure space is the space of images. We specify desired measures with landscape images from LHQ (Skorokhodov et al., 2021). Thus, DMS finds images depicting what a hiker might look like in each landscape: hikers in thick jackets for the mountains or lighter clothing for the beach, and even a baby bundled up for the snow. Each hiker is shown to the left of their corresponding landscape.

Our contributions are as follows: **(1)** We propose Discount Model Search (DMS), which improves exploration in high-dimensional measure spaces by searching over a smooth representation of the discount function (Sec. 5). **(2)** We benchmark DMS on standard QD domains (Sec. 6.1), showing it outperforms CMA-MAE and other black-box QD algorithms (Sec. 7). **(3)** We propose the QDDM setting and introduce two QDDM domains where images form the measure space (Sec. 6.2), showing that DMS also outperforms existing algorithms in these domains (Sec. 7). Overall, given the ubiquity of datasets in machine learning, we are excited for applications that can be framed as QDDM problems and tackled with DMS.

## 2 PROBLEM FORMULATION

As formulated in Fontaine & Nikolaidis (2021), black-box quality diversity (QD) optimization considers a scalar-valued *objective* function $f : \mathbb{R}^n \to \mathbb{R}$ and a vector-valued *measure* function $\boldsymbol{m} : \mathbb{R}^n \to \mathbb{R}^k$. Both functions take as input a solution $\boldsymbol{\theta} \in \mathbb{R}^n$, and $\boldsymbol{m}$ outputs $k$ measures. The image of $\boldsymbol{m}$ forms the *measure space* $S$. The *QD objective* is to find, for every $\boldsymbol{s} \in S$, a solution $\boldsymbol{\theta}$ such that $\boldsymbol{m}(\boldsymbol{\theta}) = \boldsymbol{s}$ and $f(\boldsymbol{\theta})$ is maximized. As stated, this QD objective requires infinite memory since $S$ is a continuous space. Hence, algorithms based on MAP-Elites (Mouret & Clune, 2015) discretize $S$ into a *tessellation* $T$ of $M$ *cells*, leading to a relaxed QD objective $\max_{\boldsymbol{\theta}_{1..M}} \sum_{i=1}^{M} f(\boldsymbol{\theta}_i)$. Each solution $\boldsymbol{\theta}_i$ has measures located in the region of measure space indicated by cell $i$ in $T$, and the set of solutions $\boldsymbol{\theta}_{1..M}$ is referred to as an *archive* $\mathcal{A}$.

## 3 BACKGROUND

Our work builds on several black-box QD algorithms. We include further related work in QD and image generation in Appendix A.

**MAP-Elites** (Mouret & Clune, 2015) produces a "grid archive" where the tessellation $T$ divides the measure space into a grid of axis-aligned (hyper-)rectangles. Each cell in the archive stores one solution. Each iteration, MAP-Elites selects an archive solution $\boldsymbol{\theta}$, mutates it to create a new solution $\boldsymbol{\theta}'$, and adds $\boldsymbol{\theta}'$ to the archive. As $\boldsymbol{\theta}'$ is added, it is assigned to a cell $e$ based on its measure values. $\boldsymbol{\theta}'$ replaces the solution in cell $e$ if it has a higher objective value. In this manner, MAP-Elites retains *elites*, i.e., the best solution found in each cell. We consider a version of MAP-Elites that mutates solutions by adding isotropic Gaussian noise, i.e., new solutions are created as $\boldsymbol{\theta}' \leftarrow \boldsymbol{\theta} + \sigma \mathcal{N}(\mathbf{0}, \boldsymbol{I})$.

Since grid archives require exponentially more memory in high-dimensional measure spaces, prior work (Vassiliades et al., 2018) proposes defining the tessellation $T$ as a centroidal Voronoi tessellation (CVT), where a number of centroids (e.g., 10,000) divide the measure space into equally-sized Voronoi cells. We use these *CVT archives* in our experiments in high-dimensional measure spaces.

**MAP-Elites (line)** (Vassiliades & Mouret, 2018) augments MAP-Elites with the Iso+LineDD operator, which leverages correlations between solutions in the archive by generating new solutions as $\boldsymbol{\theta}' \leftarrow \boldsymbol{\theta}_1 + \sigma_1 \mathcal{N}(\mathbf{0}, \boldsymbol{I}) + \sigma_2 \mathcal{N}(\mathbf{0}, \boldsymbol{I})(\boldsymbol{\theta}_2 - \boldsymbol{\theta}_1)$. $\boldsymbol{\theta}_1$ and $\boldsymbol{\theta}_2$ are sampled from the archive.

**Covariance Matrix Adaptation MAP-Annealing (CMA-MAE)** (Fontaine & Nikolaidis, 2023) is a state-of-the-art black-box QD algorithm that integrates the CMA-ES (Hansen, 2016) optimizer into MAP-Elites to directly optimize the QD objective (Sec. 2). CMA-MAE maintains a CMA-ES instance that samples solutions $\boldsymbol{\theta}_i$ from a Gaussian distribution $\mathcal{N}(\boldsymbol{\theta}^*, \boldsymbol{\Sigma})$. Each solution $\boldsymbol{\theta}_i$ is evaluated and added to the archive. For each $\boldsymbol{\theta}_i$, CMA-MAE computes an *improvement value* $\Delta_i$ that represents how much $\boldsymbol{\theta}_i$ improves the archive. The ranking of the improvement values enables the CMA-ES instance to update the distribution parameters $\boldsymbol{\theta}^*$ and $\boldsymbol{\Sigma}$ in the direction of greatest archive improvement; thus, CMA-ES continues to sample solutions that improve the archive in future iterations. This update causes CMA-ES to optimize the QD objective. Multiple CMA-ES instances may operate in parallel, with each instance referred to as an *emitter*.

The improvement value in CMA-MAE is defined as $\Delta(\boldsymbol{\theta}) = f(\boldsymbol{\theta}) - f_A(\boldsymbol{m}(\boldsymbol{\theta}))$, where $f_A : \mathbb{R}^k \to \mathbb{R}$ is a *discount function*. CMA-MAE represents $f_A$ as a histogram in measure space by associating a *discount value* with each cell $e$. In CMA-MAE's predecessor, CMA-ME (Fontaine et al., 2020), the discount value is the objective value of the solution $\boldsymbol{\theta}_e$ currently in the cell, i.e., $f(\boldsymbol{\theta}_e)$. When a new solution is added to cell $e$, the discount value is updated to the objective of the new solution. As in MAP-Elites, a new solution can only replace the cell solution $\boldsymbol{\theta}_e$ if it has a higher objective value.

CMA-MAE builds on the insight that CMA-ME's discount values can cause the search to quickly leave areas of the archive that require further optimization of the objective (Tjanaka et al., 2022). For example, consider if the maximum objective attainable for a cell $e$ is 100, and CMA-ME finds a solution with objective 90. Future solutions that land in $e$ garner little improvement since the discount value associated with cell $e$ is now $f(\boldsymbol{\theta}_e)$, which is 90. Even a solution $\boldsymbol{\theta}$ with objective $f(\boldsymbol{\theta}) = 100$ only receives an improvement of $100 - 90 = 10$. Thus, CMA-ME immediately searches for solutions in other areas of the archive that offer higher improvement values. In contrast, to continue optimizing for solutions that land in cell $e$, CMA-MAE sets the discount value to an *acceptance threshold* $t_e$, rather than the objective value of the solution in the cell. $t_e$ is initialized to a minimum value $f_{min}$. As a new solution $\boldsymbol{\theta}'$ enters cell $e$, $t_e$ is updated as $t_e \leftarrow (1-\alpha)t_e + \alpha f(\boldsymbol{\theta}')$, where $0 \leq \alpha \leq 1$ is an *archive learning rate* that controls how quickly $t_e$ updates. Consider a cell $e$ with $t_e = f_{min} = 0$. Given $\alpha = 0.1$, a solution with objective $f(\boldsymbol{\theta}') = 90$ updates $t_e$ as $t_e \leftarrow (1-0.1)*0 + 0.1*90 = 9$. A new solution in $e$ with objective 100 would receive improvement $100 - 9 = 81$, so CMA-MAE still receives high improvement for discovering solutions in cell $e$.

Notably, when $\alpha = 1.0$, CMA-MAE focuses on exploration and is equivalent to CMA-ME, since $t_e$ will always be set to the objective value of new solutions. When $\alpha = 0.0$, CMA-MAE performs single-objective optimization because $t_e$ will be constant, so the improvement only considers the objective, i.e., $\Delta(\boldsymbol{\theta}) = f(\boldsymbol{\theta}) - C$. Thus, adjusting the learning rate $\alpha$ enables smoothly balancing between optimization of the objective and exploration of the measure space.

**Density Descent Search (DDS)** (Lee et al., 2024) removes the discount function from CMA-MAE and introduces a kernel density estimator (KDE) that models the density of previously discovered solutions in measure space. Instead of archive improvement, DDS ranks solutions by the KDE's density estimates, prioritizing solutions in areas of low density. The KDE provides a smoother signal than CMA-MAE's discrete histogram, enabling DDS to excel at exploring measure spaces. However, since the KDE does not consider the objective, DDS does not optimize the objective, making it a *diversity optimization* algorithm, i.e., it only searches for solutions with diverse measures. Nevertheless, we draw inspiration from how the smooth signal in DDS enhances exploration.

## 4 UNDERSTANDING DISTORTION IN HIGH-DIMENSIONAL MEASURE SPACES

Distortion in QD refers to when large areas of solution space map to a small region of measure space (Fontaine & Nikolaidis, 2023). When CMA-MAE encounters distortion, the solutions it samples land in fewer archive cells since they have similar measures. Fig. 1a shows one scenario where solutions that land in the same cell interfere with CMA-MAE's improvement mechanism. On a flat (constant) objective like the one in the figure, CMA-MAE's histogram represents how often it has visited each area of measure space, and the improvement ranking guides it towards areas that it has

not visited before. For instance, Cell 3 has the lowest discount value among the three cells since it has not been explored yet, so the direction of greatest archive improvement is to generate solutions that land in Cell 3. However, since $\boldsymbol{\theta}_1$ and $\boldsymbol{\theta}_2$ both land in Cell 2, they have the same discount value. Since they also have the same constant objective, they receive the same improvement value ($\Delta_1 = \Delta_2$). Hence, CMA-MAE cannot identify the direction of greatest archive improvement.

In Fig. 1c, we present an experiment that illustrates how distortion causes solutions to land in the same cell, which activates failures like the one above. We run CMA-MAE in the 2D and 10D LP (Sphere) benchmarks, which are designed to exhibit distortion (Sec. 6.1). 2D and 10D indicate the measure space dimensionality. We plot the number of unique archive cells where solutions sampled by CMA-MAE land according to their measures; CMA-MAE samples 540 solutions per iteration. The plot shows that in both benchmarks, CMA-MAE begins searching in areas of low distortion, as many solutions land in unique cells. Over time, solutions more often land in the same cell (i.e., the number of unique cells goes down), indicating CMA-MAE has reached areas with higher distortion.

While low-dimensional measure spaces can exhibit distortion (Lee et al., 2024), our experiment shows how higher dimensions can amplify its effects. To elaborate, in Fig. 1c, the number of unique cells falls to only 30 in the higher-dimensional 10D LP (Sphere). In part, this occurs because although both the 2D and 10D benchmarks have archives with 10,000 cells, the cells in the 10D domain are exponentially larger by nature of being higher-dimensional. As such, there is a larger area of measure space where solutions sampled by CMA-MAE can fall and still be assigned the same discount value, leading to inaccurate improvement values that stall the search.

Since large cells can amplify the effects of distortion, Fontaine & Nikolaidis (2023) suggest increasing the archive resolution (i.e., adding more cells), albeit only in a 2D measure space. With higher resolution, cells are smaller, so solutions with similar measures can still fall in different cells and receive different discount values. However, this approach entails large amounts of memory, and this amount grows exponentially with measure space dimensionality. Since increasing the archive resolution effectively makes the histogram closer to a continuous function, we propose to eliminate the histogram entirely and instead search with a continuous representation of the discount function.

## 5    DISCOUNT MODEL SEARCH

To improve exploration in domains with distorted, high-dimensional measure spaces, we propose Discount Model Search (DMS). DMS trains a *discount model* to provide a smooth, continuous representation of the discount function. The key insight of DMS is that such a representation provides distinct discount values and hence improvement values, even when solutions have similar measures, making it easier to guide search towards solutions that improve the archive. For example, in Fig. 1b, the higher improvement $\Delta_2$ correctly indicates that generating solutions in the direction of $\boldsymbol{\theta}_2$ would create greater archive improvement, as such solutions would land in Cell 3, which currently has a low discount value. Below we describe DMS's components, with pseudocode shown in Algorithm 1.

**Archive and Emitters.** DMS maintains a MAP-Elites-style archive that retains the best solutions found (line 14-16), and CMA-ES-based (Hansen, 2016) emitters that optimize for archive improvement. Unlike CMA-MAE, DMS does not store discount or threshold values in the archive.

**Discount Model.** The primary component of DMS is its discount model, which approximates the true discount function $f_A : \mathbb{R}^k \to \mathbb{R}$. The discount model is a neural network $\hat{f}_A(\cdot; \psi)$ parameterized by $\psi$. It takes measures as input and outputs scalar discount values. While alternative models like kernel-based methods (Chen, 2017) are feasible, we select neural networks because the inductive biases of their various architectures make them suitable for many types of measures. For example, if the measures are low-dimensional vectors as is common in QD, an MLP (Multi-Layer Perceptron) would suffice. If the measure space includes high-dimensional data like images or text, a convolutional network (He et al., 2016) or transformer (Vaswani et al., 2017) would be suitable.

DMS trains the discount model as follows. First, to reflect that the archive is initially empty, DMS regresses the discount model to output the minimum objective $f_{min}$ at the centers of $n_{init}$ cells sampled from the archive (line 3). In the main loop (line 5-24), DMS regresses the discount model to match a dataset $\mathcal{D}_A$ of input measure values $\boldsymbol{s}$ and their corresponding discount value targets $t_A$. The dataset entries $(\boldsymbol{s}, t_A)$ come from two sources. The first source is solutions sampled by the

---

**Algorithm 1:** Discount Model Search (DMS)

---

1 **Discount Model Search** ($eval, \boldsymbol{\theta}_0, N, W, \lambda, \sigma, n_{init}, \alpha, n_{empty}, f_{min}$):

  **Input:** $eval$ function that computes objective $f$ and measures $\boldsymbol{m}$, initial solution $\boldsymbol{\theta}_0$,
    iterations $N$, num. emitters $W$, batch size $\lambda$, initial step size $\sigma$, initial training points
    $n_{init}$, archive learning rate $\alpha$, num. empty points $n_{empty}$, minimum objective $f_{min}$

  **Result:** Generates $NW\lambda$ solutions, storing elites in an archive $\mathcal{A}$

2   Initialize empty archive $\mathcal{A}$; discount model $\hat{f}_A(\cdot; \psi)$ with parameters $\psi$

3   Sample $n_{init}$ cells from $\mathcal{A}$ and regress $\hat{f}_A$ to output $f_{min}$ at the centers of these cells

4   Initialize $W$ emitters, each with mean $\boldsymbol{\theta}^* \leftarrow \boldsymbol{\theta}_0$, covariance $\boldsymbol{\Sigma} \leftarrow \sigma\boldsymbol{I}$, internal parameters $\boldsymbol{p}$

5   **for** $iter \leftarrow 1..N$ **do**

6     $\mathcal{D}_A \leftarrow []$                                    *// Dataset of measures and discount value targets.*

7     **for** *Emitter* 1 .. *Emitter* $W$ **do**

8       **for** $i \leftarrow 1..\lambda$ **do**

9         $\boldsymbol{\theta}_i \sim \mathcal{N}(\boldsymbol{\theta}^*, \boldsymbol{\Sigma})$

10        $f(\boldsymbol{\theta}_i), \boldsymbol{m}(\boldsymbol{\theta}_i) \leftarrow eval(\boldsymbol{\theta}_i)$

11        $\Delta_i \leftarrow f(\boldsymbol{\theta}_i) - \hat{f}_A(\boldsymbol{m}(\boldsymbol{\theta}_i))$   *// Compute improvement based on discount model.*

12        Compute $t_{A,i}$ with Eq. 1, where $\boldsymbol{s} = \boldsymbol{m}(\boldsymbol{\theta}_i)$

13        $\mathcal{D}_A$.append$((\boldsymbol{m}(\boldsymbol{\theta}_i), t_{A,i}))$

14        $e \leftarrow$ calculate_archive_cell$(\mathcal{A}, \boldsymbol{m}(\boldsymbol{\theta}_i))$

15        **if** $f(\boldsymbol{\theta}_i) > f(\boldsymbol{\theta}_e)$ **then**

16          Replace $\boldsymbol{\theta}_e$ (the solution in cell $e$) with $\boldsymbol{\theta}_i$

17      Rank $\boldsymbol{\theta}_i$ by $\Delta_i$

18      Adapt $\boldsymbol{\theta}^*, \boldsymbol{\Sigma}, \boldsymbol{p}$ based on improvement ranking $\Delta_i$

19      **if** *CMA-ES converges* **then**

20        Restart emitter with $\boldsymbol{\theta}^* \leftarrow$ a random solution from $\mathcal{A}$, $\boldsymbol{\Sigma} \leftarrow \sigma\boldsymbol{I}$, new internal
          parameters $\boldsymbol{p}$

21    Sample $n_{empty}$ unoccupied cells $e_{1..n_{empty}}$ from archive $\mathcal{A}$ without replacement

22    Compute centers $\boldsymbol{s}_{1..n_{empty}}$ of cells $e_{1..n_{empty}}$

23    $\mathcal{D}_A$.extend$((\boldsymbol{s}_{1..n_{empty}}, f_{min}))$

24    Regress $\hat{f}_A$ on $\mathcal{D}_A$

---

emitters (line 12-13). For each emitter solution $\boldsymbol{\theta}$, DMS creates an entry with the solution's measure values $\boldsymbol{s} = \boldsymbol{m}(\boldsymbol{\theta})$ and a target $t_A$ that reproduces CMA-MAE's threshold update rule (Sec. 3):

$$
t_A = \begin{cases} \hat{f}_A(\boldsymbol{s}) & \text{if } f(\boldsymbol{\theta}) \leq \hat{f}_A(\boldsymbol{s}) \\ (1-\alpha)\hat{f}_A(\boldsymbol{s}) + \alpha f(\boldsymbol{\theta}) & \text{if } f(\boldsymbol{\theta}) > \hat{f}_A(\boldsymbol{s}) \end{cases}
\tag{1}
$$

Here, if the objective value $f(\boldsymbol{\theta})$ is worse than its discount value $\hat{f}_A(\boldsymbol{s})$, $t_A$ is set to the current discount value. If the objective value exceeds the discount value, $t_A$ is a linear combination between the objective value and the current discount value, weighted by archive learning rate $\alpha$. Similar to CMA-MAE's threshold update rule, this target aims to slowly increase the discount values (and hence decrease the improvement values) in areas of the measure space that DMS has explored, so that the emitters are required to find solutions in new areas of measure space.

The second source of data for $\mathcal{D}_A$ is "empty points," i.e., the centers of unoccupied archive cells. During preliminary experiments, we noticed that updating the discount model caused it to change its outputs in areas of the measure space that were not represented in the dataset $\mathcal{D}_A$. In particular, in areas that had not been explored yet, the discount model should have output the minimum objective $f_{min}$, but it instead output high arbitrary values. To prevent this issue by "clamping down" the discount model in unexplored areas of the archive, we sample $n_{empty}$ unoccupied cells from the archive (line 21). We then add the center of each cell to the dataset $\mathcal{D}_A$, with an associated target of $t_A = f_{min}$ (line 22-23). If there are fewer than $n_{empty}$ unoccupied archive cells, we select all such cells. Note that in the CVT archive, the "center" of the cell is that cell's centroid.

**Summary.** DMS performs two phases. First, it searches for solutions that improve the archive $\mathcal{A}$ by sampling solutions with the emitters. Since the emitters contain CMA-ES instances, solutions are sampled from a Gaussian (line 9). As DMS progresses, each emitter updates its Gaussian based on archive improvement rankings (line 18) computed via the discount model (line 11). If CMA-ES converges, the emitters reset (line 19-20). Second, DMS trains the discount model $\hat{f}_A$, ensuring improvement values remain accurate. It collects the dataset $\mathcal{D}_A$ (line 12-13, 21-23) and regresses $\hat{f}_A$ to match these values (line 24). Thus, $\hat{f}_A$ guides the emitters to fill a high-performing archive.

## 6 Domains

We evaluate DMS on standard QD benchmarks and in a setting we refer to as Quality Diversity with Datasets of Measures (QDDM). We provide further details of all domains in Appendix E.

### 6.1 Benchmarks

**Linear Projection (LP)** (Fontaine et al., 2020) benchmarks distortion by creating a measure function that projects the majority of an $n$-dimensional solution space into the center of a $k$-dimensional measure space. We set $n = 100$ and instantiate LP with the Sphere, Rastrigin, and Flat objectives. With the Flat objective (Lee et al., 2024), which only outputs 1.0, LP becomes a *diversity optimization* domain where solutions differ only by their measures. To provide a range of objective functions and measure space dimensionalities, we use eight instantiations of LP, named by the measure space dimensionality $k$ and the objective function: **2D LP (Sphere)**, **10D LP (Sphere)**, **20D LP (Sphere)**, **50D LP (Sphere)**, **2D LP (Rastrigin)**, **10D LP (Rastrigin)**, **2D LP (Flat)**, **10D LP (Flat)**.

**Arm Repertoire** (Vassiliades & Mouret, 2018) is an inverse kinematics domain where solutions are joint angle configurations of a 2D planar arm with $n = 100$ joints. The measure function outputs the 2D position of the arm's end effector. The objective indicates the variance of the $n$ joint angles.

### 6.2 Quality Diversity with Datasets of Measures

We propose the QDDM setting, where instead of designing measure functions, a user provides a dataset indicating their desired measure values. The defining feature of QDDM is that the user provides high-dimensional data, e.g., images, audio, or text, and the measure space $S$ is the space of such data, e.g., $S \subseteq \mathbb{R}^{k=256*256*3=196,608}$ if the data are $256 \times 256 \times 3$ RGB images.

Initially, it seems problematic to construct an archive for QDDM due to the high dimensionality of the measure space. However, we adopt the manifold hypothesis (Fefferman et al., 2016), i.e., the assumption that most high-dimensional data lie on a low-dimensional manifold embedded within the high-dimensional space. We recognize that the distribution of measures relevant to a user occupies only a small region of the overall measure space, and this distribution is reflected in the user's dataset. Hence, we propose to construct a CVT archive where the centroids are the points in the dataset. Here, the CVT no longer uniformly partitions the measure space. Rather, *it only partitions the small region of measure space desired by the user and indicated in the dataset.* However, the CVT archive introduces a new consideration for QDDM, viz., the choice of *distance function*. To locate the cell where a solution belongs, CVT archives find the centroid closest to the solution's measures. While Euclidean distance is common here, it is not always ideal (Vassiliades et al., 2018), which may be especially true when the measures are as high-dimensional as images or text.

**Triangle Arrangement (TA).** We introduce two QDDM domains. First, TA builds on computational creativity domains (Tian & Ha, 2022) and involves arranging a prespecified number of triangles to create images. A solution consists of the vertices, brightness (we consider grayscale images), and alpha (transparency) of each triangle. A solution's measure is created by rendering the triangles as a raster image. We specify desired images (measures) by sampling 1000 images from either MNIST (Lecun et al., 1998) or Fashion MNIST (Xiao et al., 2017), leading to two versions of this domain: **TA (MNIST)** and **TA (F-MNIST)**. Since the images in these datasets are $28 \times 28$, the measure space is 784-dimensional. Drawing from the loss function in Tian & Ha (2022), we define the distance function as Euclidean distance, so each solution is placed in the archive cell of the digit it most resembles. To make the triangle images resemble the MNIST images, we define the objective

Table 1: Mean QD Score ("QD") and Coverage ("Cov") for each algorithm in each domain.

| | 2D LP (Sphere) | | 10D LP (Sphere) | | 20D LP (Sphere) | | 50D LP (Sphere) | |
|---|---|---|---|---|---|---|---|---|
| | QD | Cov | QD | Cov | QD | Cov | QD | Cov |
| DMS | **6,978.20** | **95.89%** | **6,409.50** | **89.21%** | **7,406.01** | **95.97%** | **6,991.00** | **87.00%** |
| CMA-MAE | 6,327.90 | 80.95% | 608.53 | 6.95% | 881.76 | 9.13% | 2,327.11 | 24.21% |
| DDS | 3,156.24 | 70.75% | 4,237.72 | 60.07% | 6,990.51 | 84.31% | 6,373.24 | 74.30% |
| MAP-Elites (line) | 4,908.81 | 60.42% | 2,570.74 | 29.20% | 5,936.66 | 65.86% | 5,204.80 | 55.77% |
| MAP-Elites | 4,163.41 | 50.76% | 228.65 | 2.35% | 3,280.34 | 35.62% | 3,870.48 | 40.89% |

| | 2D LP (Rastrigin) | | 10D LP (Rastrigin) | | 2D LP (Flat) | | 10D LP (Flat) | |
|---|---|---|---|---|---|---|---|---|
| | QD | Cov | QD | Cov | QD | Cov | QD | Cov |
| DMS | **5,738.90** | **91.67%** | **5,138.81** | **88.19%** | **7,902.05** | **79.02%** | **7,982.15** | **79.82%** |
| CMA-MAE | 5,258.59 | 80.14% | 246.55 | 2.98% | 5,675.90 | 56.76% | 1,554.90 | 15.55% |
| DDS | 2,495.11 | 71.68% | 3,331.70 | 59.54% | 6,967.75 | 69.68% | 6,004.95 | 60.05% |
| MAP-Elites (line) | 3,841.05 | 56.63% | 2,001.76 | 28.04% | 4,510.65 | 45.11% | 757.75 | 7.58% |
| MAP-Elites | 3,172.59 | 48.21% | 499.66 | 7.09% | 4,327.00 | 43.27% | 125.65 | 1.26% |

| | Arm Repertoire | | TA (MNIST) | | TA (F-MNIST) | | LSI (Hiker) | |
|---|---|---|---|---|---|---|---|---|
| | QD | Cov | QD | Cov | QD | Cov | QD | Cov |
| DMS | **7,963.44** | 80.15% | 951.56 | **99.84%** | **701.14** | **72.28%** | 214.91 | 3.77% |
| CMA-MAE | 7,902.43 | 79.22% | **954.27** | 99.48% | 625.65 | 63.92% | 14.61 | 1.56% |
| DDS | 5,568.23 | **80.24%** | — | — | — | — | — | — |
| MAP-Elites (line) | 7,458.67 | 75.60% | 945.60 | 98.86% | 551.13 | 56.68% | -51,827.44 | **7.49%** |
| MAP-Elites | 7,411.10 | 75.42% | 941.94 | 98.42% | 513.13 | 52.68% | -18,917.87 | 5.06% |

as the negative (to facilitate maximization) mean squared error between the triangle image and the archive centroid (MNIST image) to which it is assigned.

**Latent Space Illumination (LSI).** LSI entails exploring the latent space of a generative model to create images with diverse measures. While prior work (Fontaine et al., 2021b) considers LSI with low-dimensional measures, we consider LSI where the measures are images. As described in Sec. 1, we search for images of hikers in different landscapes. The solutions are latent vectors ($w$-space, but not $w^+$-space) of StyleGAN3 (Karras et al., 2021a), with images of faces output by StyleGAN3 serving as measures. Thus, the measure space is the space of $256 \times 256 \times 3$ images. The desired measures are specified with a dataset of 10,000 landscape images sampled from LHQ256 (Skorokhodov et al., 2021). To associate hikers with landscapes in the CVT archive, the distance function is the CLIP score (Radford et al., 2021) between the face and landscape; CLIP score is more semantically meaningful than Euclidean distance. The objective is the CLIP score between the face image and the prompt "A photo of the face of a hiker." We also add a regularizer loss (Fontaine & Nikolaidis, 2023) to penalize latent vectors that fall outside the StyleGAN3 training distribution, and similar to TA, we reward higher alignment between faces and landscapes by adding the CLIP score between the face and the landscape to which it is assigned. We refer to this domain as **LSI (Hiker)**.

## 7 EXPERIMENTS

We evaluate DMS in low- and high-dimensional measure spaces through experiments in 9 benchmarks [2D LP (Sphere), 10D LP (Sphere), 20D LP (Sphere), 50D LP (Sphere), 2D LP (Rastrigin), 10D LP (Rastrigin), 2D LP (Flat), 10D LP (Flat), Arm Repertoire] and 3 QDDM domains [TA (MNIST), TA (F-MNIST), LSI (Hiker)]. In each domain, we conduct a between-groups study with the algorithm as the independent variable: besides DMS, we consider the black-box QD algorithms described in Sec. 3: CMA-MAE, DDS, MAP-Elites (line), and MAP-Elites. We consider two dependent variables. *QD Score* (Pugh et al., 2016) represents overall performance by summing the objectives of all solutions in the archive, as is done in the QD objective (Sec. 2). *Coverage* indicates how much of the measure space has been explored by computing the percentage of archive cells that have a solution in them. Note that the objective is normalized to have a maximum value of 1 in all domains. *Our hypothesis is that DMS will outperform all other algorithms in both QD Score and Coverage.* We implement all algorithms with pyribs (Tjanaka et al., 2023); hyperparameters and further details are in Appendix C and F.

## 7.1 ANALYSIS

Table 1 shows the results from 20 trials in the benchmark domains and 5 trials in the QDDM domains. Fig. 2 shows sample images from DMS in LSI (Hiker). Refer to Appendix B for performance plots, error bars, archive heatmaps, and discount function visualizations. We could not run DDS in the QDDM domains due to the KDE's runtime, which grows linearly with measure space dimensionality. Visual inspection showed the results were normally distributed, but Levene's test showed most settings were non-homoscedastic. Thus, to analyze the results, we conducted Welch's one-way ANOVA in each domain for each dependent variable. All ANOVAs were significant ($p < 0.001$; $F$ values in Appendix B), so we performed pairwise comparisons with the Games-Howell test.

**Benchmark Domains.** In the benchmarks, DMS significantly outperformed all baselines in QD Score and Coverage, except in Arm Repertoire, where DDS had significantly better coverage. The high performance on LP shows that DMS better overcomes distortion than previous algorithms, as these domains are designed to benchmark distortion. Since DDS is a diversity optimization algorithm, we expect it to achieve the best coverage in all domains, so it is surprising that DMS outperforms it in almost all domains, even 2D and 10D LP (Flat), which are diversity optimization domains where the objective is always 1.0.

**TA Domains.** In TA (MNIST), for both metrics, DMS significantly outperformed the two MAP-Elites algorithms, but there was no significant difference with CMA-MAE. In TA (F-MNIST), DMS significantly outperformed all baselines in both metrics. The coverage results in TA (MNIST) illustrate that not all QDDM domains exhibit high distortion — low distortion is reflected in how all algorithms achieve nearly perfect coverage in TA (MNIST). We believe the high coverage stems from how MNIST images are fairly similar in appearance. As such, an algorithm can fill the archive by generating triangle images that differ only slightly from one another. TA (F-MNIST) seems more challenging, as no algorithm achieves perfect coverage there.

On the other hand, the difficulty of TA (MNIST) seems to lie in optimization of the objective, as the difference in QD Score between DMS/CMA-MAE and the two MAP-Elites algorithms is small yet statistically significant. This property suggests a potential limitation of DMS that may be suitable for exploration in future work. We speculate that by nature of being a model, the discount model in DMS exhibits errors, which we can imagine as adding noise to discount values. In domains where objective optimization is less important, we hypothesize that the noise is small enough that improvement rankings remain unaffected. However, in a domain that requires fine optimization of the objective like TA (MNIST), this noise interferes with improvement rankings, hindering DMS. CMA-MAE maintains exact values in its histogram and would not have such noise, potentially explaining why DMS does not outperform CMA-MAE's QD Score here.

**LSI (Hiker).** The results in LSI (Hiker) highlight the difficulty of complex QDDM domains. Here, DMS significantly outperformed CMA-MAE in both metrics, but there was no significant difference with the two MAP-Elites algorithms. While DMS outperforms CMA-MAE, it covers only 3.77% of the archive, although this still represents 377 hiker images. We note that the two MAP-Elites algorithms receive large negative QD Scores and high coverages by generating latent vectors far outside the training distribution of StyleGAN3 and incurring large regularization losses. Similarly, they exhibit high performance variance, so they have no significant difference with DMS.

**Ablation Study.** We ablate the hyperparameters of DMS in Appendix D. We find that the archive learning rate $\alpha$ behaves similarly as in CMA-MAE: intermediate values enable DMS to balance optimization of the objective and exploration of the measure space, while $\alpha = 0$ causes DMS to over-emphasize the objective. Meanwhile, the "empty points" are necessary for training the discount model: removing them by setting $n_{empty} = 0$ causes performance to drop since the discount model takes on arbitrary values in areas of the measure space that have not been explored (Fig. 11). In contrast, setting $n_{empty} = 10, 100,$ or $1000$ resolves this issue by "clamping down" the discount model (Fig. 8). Overall, our experimental results show how the discount model successfully guides optimization in DMS, leading to high performance across domains with different measure spaces.

**Computation Time.** Since DMS trains a discount model, it may be more computationally expensive than other methods like CMA-MAE. To quantify this tradeoff, we recorded the wallclock time when running all the QD algorithms in our experiments. These results are summarized in Table 2. We focus on comparing DMS and CMA-MAE, as their primary difference is in training the discount model. On the benchmark domains (LP and Arm Repertoire), DMS takes noticeably longer than

Table 2: Computation (wallclock) time (in seconds) of each algorithm in each domain. We show the mean and standard error of the mean over 20 trials for the benchmark domains and 5 trials for the QDDM domains.

| | 2D LP (Sphere) | 10D LP (Sphere) | 20D LP (Sphere) | 50D LP (Sphere) |
|---|---|---|---|---|
| DMS | 397.83 ±0.37 | 876.33 ±44.22 | 2,797.61 ±6.57 | 7,843.98 ±228.28 |
| CMA-MAE | 121.13 ±0.10 | 333.92 ±1.70 | 1,316.87 ±22.32 | 4,707.92 ±86.53 |
| DDS | 1,918.68 ±22.42 | 3,535.78 ±1.53 | 1,481.18 ±30.98 | **2,600.24 ±78.20** |
| MAP-Elites (line) | **31.10 ±0.10** | 231.50 ±0.73 | 883.23 ±1.04 | 4,147.29 ±102.89 |
| MAP-Elites | 39.01 ±0.21 | **190.25 ±2.46** | **873.06 ±0.35** | 4,523.16 ±51.12 |

| | 2D LP (Rastrigin) | 10D LP (Rastrigin) | 2D LP (Flat) | 10D LP (Flat) |
|---|---|---|---|---|
| DMS | 707.20 ±21.31 | 711.08 ±30.76 | 683.53 ±27.73 | 723.34 ±32.31 |
| CMA-MAE | 182.07 ±0.36 | 465.80 ±3.81 | 165.33 ±0.25 | 800.12 ±0.92 |
| DDS | 1,957.29 ±0.87 | 3,598.86 ±2.56 | 1,937.41 ±1.51 | 3,581.79 ±2.05 |
| MAP-Elites (line) | **39.74 ±0.14** | **274.86 ±0.62** | **30.84 ±0.09** | **339.73 ±0.42** |
| MAP-Elites | 51.27 ±0.17 | 367.56 ±1.57 | 38.41 ±0.14 | 451.46 ±0.41 |

| | Arm Repertoire | TA (MNIST) | TA (F-MNIST) | LSI (Hiker) |
|---|---|---|---|---|
| DMS | 687.85 ±27.18 | 489.90 ±12.55 | 546.60 ±21.46 | 4,740.24 ±482.74 |
| CMA-MAE | 130.83 ±0.26 | 495.95 ±13.12 | 525.42 ±19.22 | 4,690.12 ±654.47 |
| DDS | 1,577.54 ±1.70 | — | — | — |
| MAP-Elites (line) | 39.68 ±0.10 | **247.21 ±4.07** | **231.91 ±0.38** | 3,470.08 ±12.88 |
| MAP-Elites | **32.80 ±0.08** | 291.98 ±1.46 | 309.01 ±3.92 | **3,460.71 ±5.86** |

CMA-MAE due to this training. However, on the QDDM domains (TA and LSI), this difference is much less noticeable, because the evaluation of solutions becomes the bottleneck, rather than the QD algorithm itself. Meanwhile, we note that MAP-Elites and MAP-Elites (line) are the fastest algorithms because their search operations are the least computationally expensive, since they are rooted in fixed Gaussian noise. In contrast, DMS and CMA-MAE manipulate multivariate Gaussian distributions in their CMA-ES emitters.

## 8 CONCLUSION

By searching in distorted and high-dimensional measure spaces, DMS offers two benefits for QD practitioners. First, DMS can improve performance in current QD applications. As the results in the benchmark domains show (Sec. 7.1), DMS outperforms current algorithms in various domains with low-dimensional measure spaces. Since most current applications involve low-dimensional measure spaces, we believe DMS will also outperform current QD algorithms in current applications. Second, DMS enables *new* applications by addressing the proposed QDDM setup (Sec. 6.2), where diversity in a high-dimensional measure space like images is specified by providing a dataset. We believe framing measures in terms of datasets makes QD more accessible by not only alleviating the need to hand-design measure functions, but also making it possible to specify measures that cannot easily be represented by low-dimensional values. For example, as the TA and LSI (Hiker) domains showed, we can now specify the measure space in vision and art domains with image datasets. Overall, given that datasets are central to machine learning, we believe it will prove fruitful to frame problems across machine learning as QDDM problems and solve them with algorithms like DMS.

Our paper has several limitations. First, while searching over the discount model garners high performance, training it induces computational overhead (Sec. 7.1). Second, while DMS trains the discount model with targets that reproduce CMA-MAE's threshold update, alternative targets may improve properties like smoothness. Finally, we primarily consider small MLPs for the discount model. We are excited for future work in domains that require more advanced discount model architectures.

## 9 ETHICS STATEMENT

DMS has the potential for some negative societal impacts. For example, its abilities in QDDM settings may exacerbate biases in the dataset of measures, image generator (Karras et al., 2021a), and even distance metric (Radford et al., 2021). These biases can be mitigated by carefully managing all datasets, including that used to train the generator (Perera & Patel, 2023; Maluleke et al., 2022).

In general, QD can also debias models by generating balanced datasets (Chang et al., 2024). In addition, using QDDM methods to drive generative search towards specific artistic styles defined in the measure space can discount unique styles created by human artists (Shan et al., 2023). Alternative objective functions (Tseng et al., 2021) may enable finding solutions that reflect the dataset of measures without mimicking artists' creativity.

## 10 REPRODUCIBILITY STATEMENT

We have included details of DMS in Sec. 5, including pseudocode in Algorithm 1. We include details of our domains and experimental setup in Sec. 6 and Sec. 7, along with hyperparameters in Appendix C, extended domain details in Appendix E, and further implementation details in Appendix F. We have released our code at `https://github.com/icaros-usc/discount-models`. Inside the code repository, a README file lists instructions for setting up the environment and reproducing all experiments in this paper.

### ACKNOWLEDGMENTS

We thank Varun Bhatt, Saeed Hedayatian, and Aaquib Tabrez for their insightful feedback. This work was partially supported by the NSF CAREER (#2145077), NSF NRI (#2024949), NSF GRFP (#DGE-1842487), and the DARPA EMHAT project.

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

## A    RELATED WORK

**Quality Diversity Optimization.**  Applications of QD include robotics (Cully et al., 2015; Huber et al., 2025; Morrison et al., 2020), drug discovery (Boige et al., 2023; Verhellen & Van den Abeele, 2020), urban planning (Galanos et al., 2021), and finance (Gašperov et al., 2025). In computer vision, QD can create diverse images by exploring the latent space of generative models, e.g., StyleGAN (Karras et al., 2021b) in our work and in (Fontaine & Nikolaidis, 2021), and Stable Diffusion (Rombach et al., 2022) in (Ding et al., 2024). Such images can form a synthetic dataset for debiasing downstream models (Chang et al., 2024). In reinforcement learning, QD can generate diverse locomotion policies (Nilsson & Cully, 2021; Pierrot et al., 2022; Batra et al., 2024; Chalumeau et al., 2023; Xue et al., 2024), and in red-teaming, QD can probe a large language model (LLM) to produce harmful outputs (Samvelyan et al., 2025).

Our work fits a growing trend of integrating models into QD. To accelerate evaluations, multiple works (Bhatt et al., 2022; 2023; Kent et al., 2025; Zhang et al., 2022; Gaier et al., 2018; Hagg et al., 2020) train surrogate models to approximate expensive objectives and/or measures. Others (Gravina et al., 2016; 2019; Salehi et al., 2021) build models that guide the creation of new solutions, especially in reinforcement learning (Tjanaka et al., 2022; Nilsson & Cully, 2021; Pierrot et al., 2022; Batra et al., 2024; Chalumeau et al., 2023; Grillotti et al., 2024). Furthermore, whereas DMS searches directly in the space of high-dimensional measures, several approaches (Cully, 2019; Paolo et al., 2020; Grillotti & Cully, 2022; McCormack & Cruz Gambardella, 2022) build models that compress high-dimensional measures into low-dimensional measures. Finally, as discussed in Sec. 6.2, we apply the manifold hypothesis (Fefferman et al., 2016) in *measure space* when creating the archive in QDDM domains. Prior work (Vassiliades & Mouret, 2018; Gaier et al., 2020; Rakicevic et al., 2021; Hegde et al., 2023) applies the manifold hypothesis in *solution space* by searching over the *elite hypervolume*, a low-dimensional manifold where the solutions to each QD problem are hypothesized to exist.

Related to our proposed QDDM setting, developing intuitive ways to specify measures of diversity is an active research area in QD. For example, QD through AI Feedback (QDAIF) (Bradley et al., 2024) evaluates measures by querying LLMs for feedback, while the LSI domain in prior work (Fontaine & Nikolaidis, 2021) computes the CLIP score between text prompts and generated images. QD through Human Feedback (QDHF) (Ding et al., 2024) learns measures from human preferences via a contrastive loss. Each method's suitability depends on which user effort is easiest. For example, QDAIF excels when measures can be conveniently specified to an LLM evaluator that outputs a vector of measures. Conversely, QDAIF can be limited by the challenges of prompt engineering and the stochasticity of LLM outputs. Moreover, it may be difficult to create vector-valued measures that elicit the desired diversity, like in LSI (Hiker), where QDDM makes it easy to specify "where a hiker is located" with landscape images. A key distinction is that QDDM specifies desired measure *values*, while the above methods all define measure *functions*. If an appropriate dataset does not exist, designing a measure function may be more appropriate, since datasets require significant effort to curate. On the other hand, we believe QDDM will be applicable in many problems since datasets are abundant in machine learning.

**Computational Creativity.** Our TA domains draw from prior work (Tian & Ha, 2022) that arranges triangles to represent images and text prompts with evolution strategies. Similarly, various works arrange basic shapes to create artistic images (Paauw & Van den Berg, 2019; Berg et al., 2019; Cason, 2016; Fogleman, 2016). To our knowledge, QD algorithms have not been applied in this setting, but they have generated other forms of art, such as line drawings (McCormack & Cruz Gambardella, 2022) and images (Zammit et al., 2022).

**Latent Space Exploration.** LSI (Hiker) is an example of *latent space illumination*, where a QD algorithm searches for latent vectors that elicit diverse, high-performing outputs of a generative model. LSI was first introduced to generate video game levels (Fontaine et al., 2021b;a; Sarkar & Cooper, 2021; Schrum et al., 2020; Steckel & Schrum, 2021) and 2D shapes (Hagg et al., 2021). Later work (Fontaine & Nikolaidis, 2021; 2023) explored the latent space of StyleGAN (Karras et al., 2021b) to generate celebrity images with low-dimensional (2D) measures based on the CLIP score (Radford et al., 2021; Herrera-Berg, 2021). Beyond LSI, various methods aim to navigate latent spaces. Several works discover interpretable directions in GAN latent spaces, where these directions manipulate pose or facial features (Shen & Zhou, 2021; Shen et al., 2020; Voynov &

Babenko, 2020; Shen et al., 2022) or transform position and scale (Plumerault et al., 2020). In single-objective optimization, prior works search the latent spaces of generative models to create video game levels (Volz et al., 2018; Merino et al., 2023; Tanabe et al., 2021) or synthetic fingerprints (Bontrager et al., 2018) that satisfy desired characteristics.

# B    RESULTS

We present further results obtained in our main experiments in Sec. 7. Fig. 3 and Fig. 4 show the mean and standard error of the mean of both dependent variables (QD Score and Coverage), for all algorithms in all domains. The plots shows the values over 10,000 iterations, and the table shows the final values. Note that since the objective is always 1.0 in the LP (Flat) domains, the QD Score and Coverage differ by a factor of the number of cells in the archive, i.e., the QD Score is 10,000 times the Coverage. In the plot for LSI (Hiker), the QD Score is initially negative since the algorithms receive a regularization penalty due to generating images outside the training distribution of StyleGAN3. Neither MAP-Elites variant's QD Score is visible due to being large negative values.

Fig. 5 and Fig. 6 show sample images from DMS in the TA (MNIST) and TA (F-MNIST) domains. Note that by default, these domains render the triangles into a $28 \times 28$ image. However, since the triangles in each solution form a vector graphic, they can be rendered at any resolution. Thus, for visualization purposes, we rendered them at $280 \times 280$ resolution in these figures.

Fig. 7 shows heatmaps of a randomly selected archive of each algorithm, in domains with 2D measure spaces. Fig. 8 shows how the archive and discount model in DMS progresses over iterations in the 2D LP (Sphere) benchmark. We include further descriptions and analyses of both of these figures in their captions.

Table 3 shows the results of Welch's one-way ANOVA for both dependent variables in all domains. Note that large between-group variances led to many large $F$ statistics. Table 4 and Table 5 show the results of pairwise comparisons between each pair of algorithms in each domain. Each entry compares the method in the row to the method in the column, with > indicating significantly greater, < indicating significantly less, − indicating no significant difference, and $\varnothing$ indicating an invalid comparison. For example, DMS significantly outperforms DDS in QD Score in 2D LP (Sphere). Note that since we were unable to run DDS in the QDDM domains [TA (MNIST), TA (F-MNIST), and LSI (Hiker)], the degrees of freedom are smaller for those domains' ANOVAs, and there is no comparison between DDS and other algorithms in the pairwise comparison tables.

## B.1    COMPARABILITY OF RESULTS TO PRIOR WORK

Variations of the linear projection and arm repertoire domains have appeared in a number of prior works, each with slightly different setups than the one in our work. Here, to facilitate better comparability with the results in prior work, we clarify the distinctions between our setup and previous ones. The first difference is in the order of magnitude of the scores reported in our work. Prior work like Fontaine 2021 (Fontaine & Nikolaidis, 2021) and Fontaine 2023 (Fontaine & Nikolaidis, 2023) reported *normalized* QD scores, which are QD scores divided by the number of cells in the archive (10,000). Those works also normalized objective values to be 0-100, whereas we normalize to 0-1 to make the objective values more amenable to the discount model. These two changes explain the difference in orders of magnitude between our results and those works: when divided by 10,000 and multiplied by 100, our results for CMA-MAE, MAP-Elites, and MAP-Elites (line) on 2D LP (Sphere), 2D LP (Rastrigin), and Arm Repertoire are nearly identical to those for LP (sphere), LP (Rastrigin), and Arm Repertoire in Fontaine 2023 (our remaining benchmark domains were not included in Fontaine 2023).

The second difference is in the number of iterations run. Like Fontaine 2023, we run all algorithms for 10,000 iterations. However, Lee 2024 (Lee et al., 2024) ran DDS and other algorithms for 5,000 iterations. As a result, our Coverage for DDS, CMA-MAE, and MAP-Elites (line) in 2D LP (Flat) and 10D LP (Flat) is higher than that in the LP and Multi-feature LP domains of Lee 2024.

The third difference is in the solution space dimensionality in the benchmarks. We follow Fontaine 2023 in considering a 100D solution space in each benchmark domain. In contrast, Fontaine 2021 considered 1000D solution spaces to emphasize the scalability of differentiable quality diversity in

high-dimensional solution spaces. Due to the difference in setting, our results are not comparable to those from Fontaine 2021.

Finally, Fontaine 2020 (Fontaine et al., 2020) considers fundamentally different hyperparameters than those in any of these works, e.g., it runs for 4500 iterations and has archives with 250,000 cells, whereas our work and most of these works run for 10,000 iterations and have archives with 10,000 cells.

Figure 3: Mean and standard error of the mean for QD Score and Coverage of each algorithm in each domain. Standard error may not be visible in some plots.

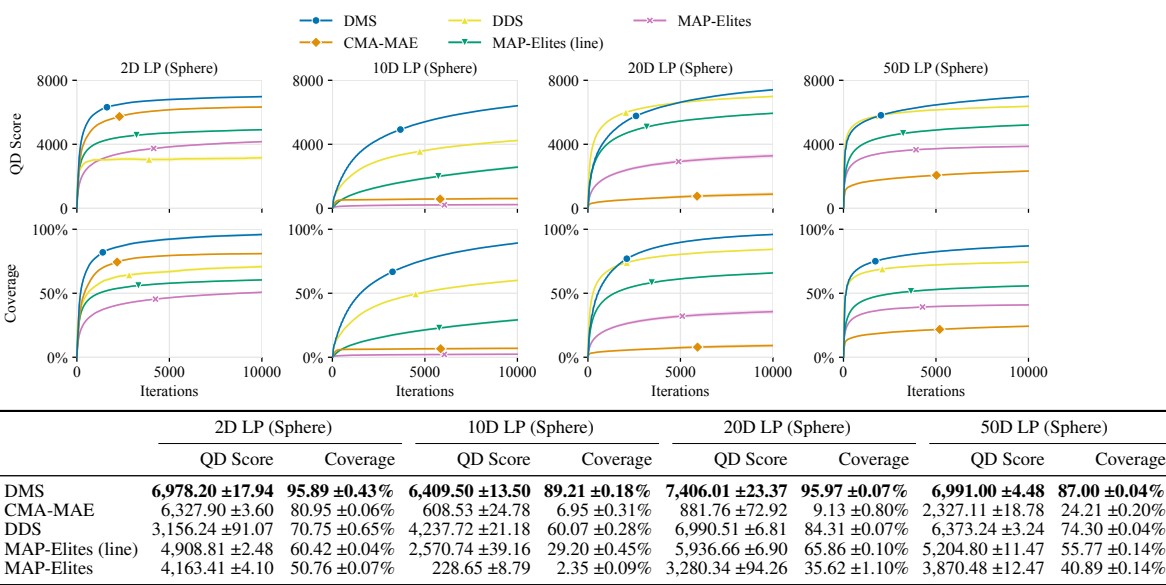

|  | 2D LP (Sphere) | | 10D LP (Sphere) | | 20D LP (Sphere) | | 50D LP (Sphere) | |
|---|---|---|---|---|---|---|---|---|
|  | QD Score | Coverage | QD Score | Coverage | QD Score | Coverage | QD Score | Coverage |
| DMS | **6,978.20 ±17.94** | **95.89 ±0.43%** | **6,409.50 ±13.50** | **89.21 ±0.18%** | **7,406.01 ±23.37** | **95.97 ±0.07%** | **6,991.00 ±4.48** | **87.00 ±0.04%** |
| CMA-MAE | 6,327.90 ±3.60 | 80.95 ±0.06% | 608.53 ±24.78 | 6.95 ±0.31% | 881.76 ±72.92 | 9.13 ±0.80% | 2,327.11 ±18.78 | 24.21 ±0.20% |
| DDS | 3,156.24 ±91.07 | 70.75 ±0.65% | 4,237.72 ±21.18 | 60.07 ±0.28% | 6,990.51 ±6.81 | 84.31 ±0.07% | 6,373.24 ±3.24 | 74.30 ±0.04% |
| MAP-Elites (line) | 4,908.81 ±2.48 | 60.42 ±0.04% | 2,570.74 ±39.16 | 29.20 ±0.45% | 5,936.66 ±6.90 | 65.86 ±0.10% | 5,204.80 ±11.47 | 55.77 ±0.14% |
| MAP-Elites | 4,163.41 ±4.10 | 50.76 ±0.07% | 228.65 ±8.79 | 2.35 ±0.09% | 3,280.34 ±94.26 | 35.62 ±1.10% | 3,870.48 ±12.47 | 40.89 ±0.14% |

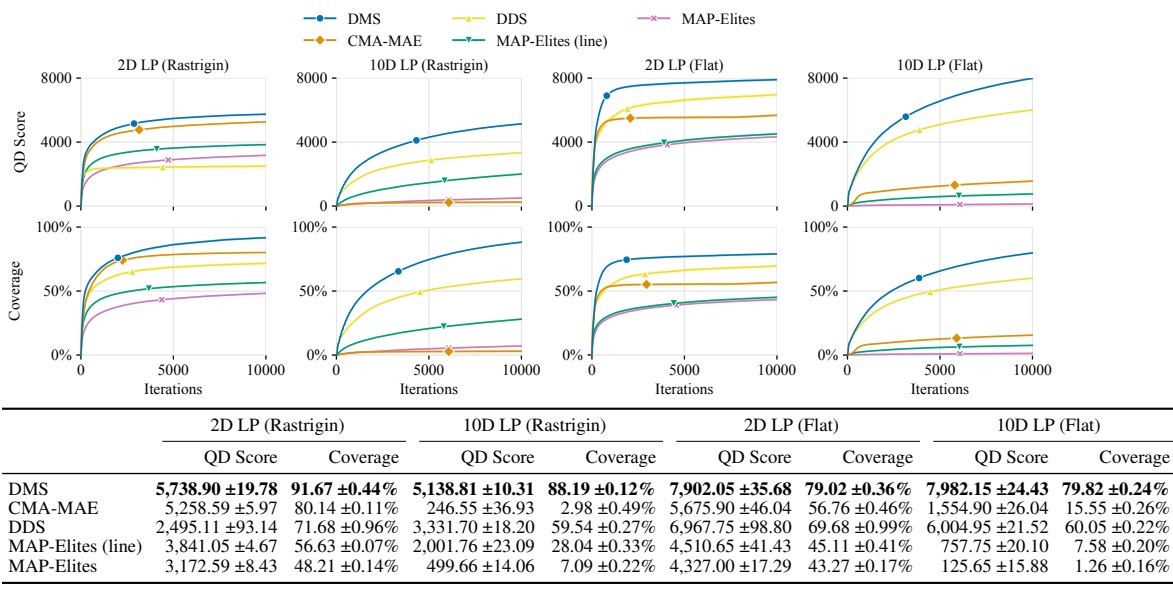

|  | 2D LP (Rastrigin) | | 10D LP (Rastrigin) | | 2D LP (Flat) | | 10D LP (Flat) | |
|---|---|---|---|---|---|---|---|---|
|  | QD Score | Coverage | QD Score | Coverage | QD Score | Coverage | QD Score | Coverage |
| DMS | **5,738.90 ±19.78** | **91.67 ±0.44%** | **5,138.81 ±10.31** | **88.19 ±0.12%** | **7,902.05 ±35.68** | **79.02 ±0.36%** | **7,982.15 ±24.43** | **79.82 ±0.24%** |
| CMA-MAE | 5,258.59 ±5.97 | 80.14 ±0.11% | 246.55 ±36.93 | 2.98 ±0.49% | 5,675.90 ±46.04 | 56.76 ±0.46% | 1,554.90 ±26.04 | 15.55 ±0.26% |
| DDS | 2,495.11 ±93.14 | 71.68 ±0.96% | 3,331.70 ±18.20 | 59.54 ±0.27% | 6,967.75 ±98.80 | 69.68 ±0.99% | 6,004.95 ±21.52 | 60.05 ±0.22% |
| MAP-Elites (line) | 3,841.05 ±4.67 | 56.63 ±0.07% | 2,001.76 ±23.09 | 28.04 ±0.33% | 4,510.65 ±41.43 | 45.11 ±0.41% | 757.75 ±20.10 | 7.58 ±0.20% |
| MAP-Elites | 3,172.59 ±8.43 | 48.21 ±0.14% | 499.66 ±14.06 | 7.09 ±0.22% | 4,327.00 ±17.29 | 43.27 ±0.17% | 125.65 ±15.88 | 1.26 ±0.16% |

Figure 4: Mean and standard error of the mean for QD Score and Coverage of each algorithm in each domain. Standard error may not be visible in some plots.

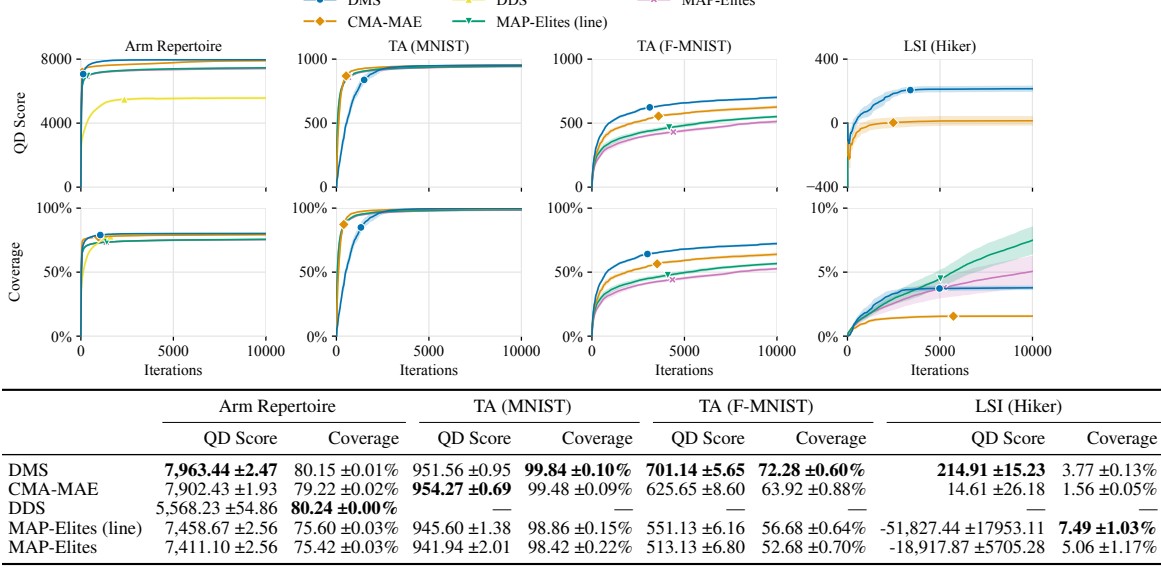

| | Arm Repertoire | | TA (MNIST) | | TA (F-MNIST) | | LSI (Hiker) | |
|---|---|---|---|---|---|---|---|---|
| | QD Score | Coverage | QD Score | Coverage | QD Score | Coverage | QD Score | Coverage |
| DMS | **7,963.44 ±2.47** | 80.15 ±0.01% | 951.56 ±0.95 | **99.84 ±0.10%** | **701.14 ±5.65** | **72.28 ±0.60%** | **214.91 ±15.23** | 3.77 ±0.13% |
| CMA-MAE | 7,902.43 ±1.93 | 79.22 ±0.02% | **954.27 ±0.69** | 99.48 ±0.09% | 625.65 ±8.60 | 63.92 ±0.88% | 14.61 ±26.18 | 1.56 ±0.05% |
| DDS | 5,568.23 ±54.86 | **80.24 ±0.00%** | — | — | — | — | — | — |
| MAP-Elites (line) | 7,458.67 ±2.56 | 75.60 ±0.03% | 945.60 ±1.38 | 98.86 ±0.15% | 551.13 ±6.16 | 56.68 ±0.64% | -51,827.44 ±17953.11 | **7.49 ±1.03%** |
| MAP-Elites | 7,411.10 ±2.56 | 75.42 ±0.03% | 941.94 ±2.01 | 98.42 ±0.22% | 513.13 ±6.80 | 52.68 ±0.70% | -18,917.87 ±5705.28 | 5.06 ±1.17% |

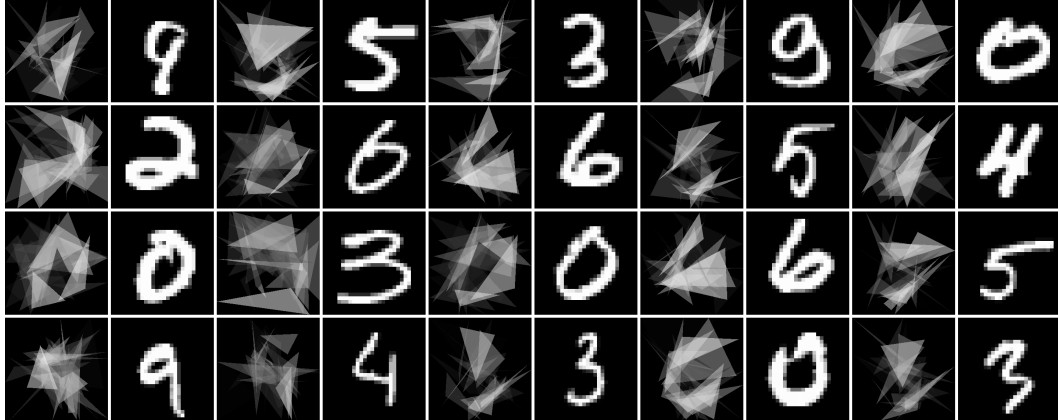

Figure 5: A random subset of images generated by DMS in the TA (MNIST) domain, where desired measures are sampled from the MNIST dataset. The goal in this domain is to arrange triangles to look like the given MNIST digits. Each rendered triangle image is shown to the left of its corresponding MNIST digit.

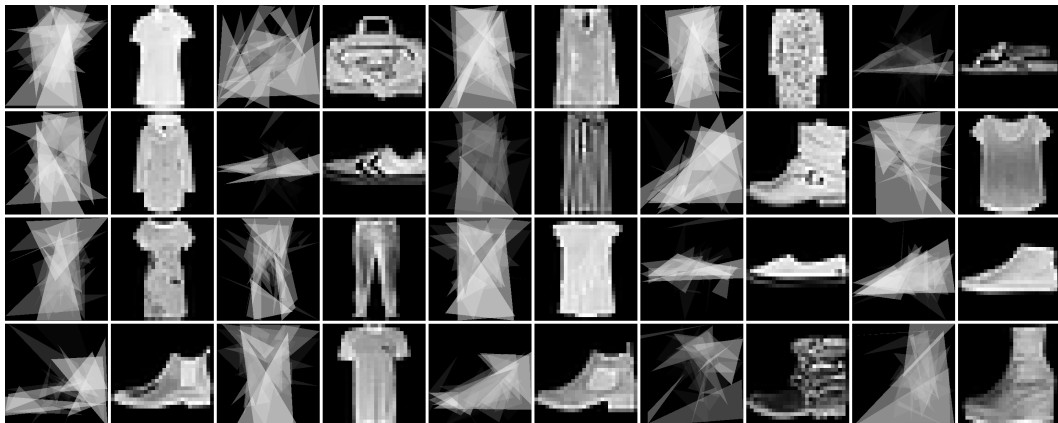

Figure 6: A random subset of images generated by DMS in the TA (F-MNIST) domain, where desired measures are sampled from the Fashion MNIST dataset. The goal in this domain is to arrange triangles to look like the images of fashion items. Each rendered triangle image is shown to the left of its corresponding fashion item.

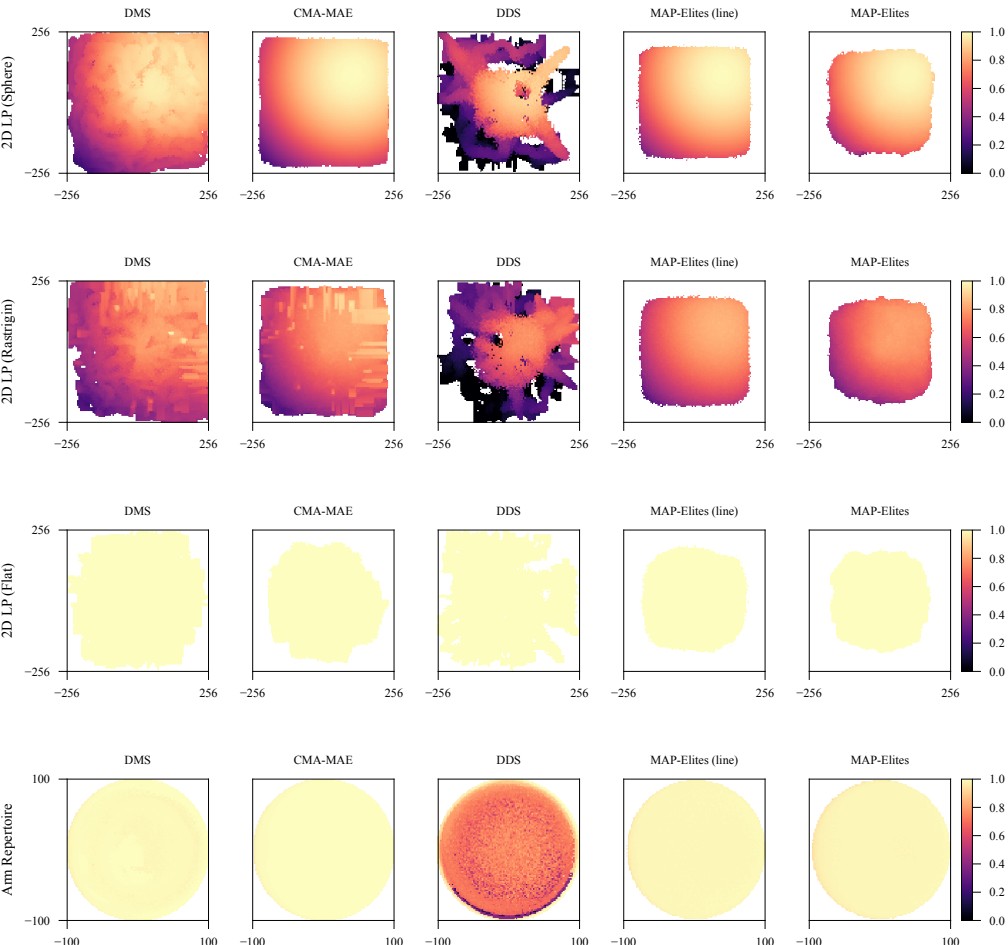

Figure 7: Heatmaps of a randomly selected archive produced by each algorithm in domains with 2D measure spaces. Each row contains heatmaps for a single domain. The axes of the heatmaps are the measures, while the color of each cell indicates the objective value. Notably, the heatmaps show how DMS achieves high coverage of the measure space. They also show how DDS achieves good coverage but cannot achieve high objective values since it is a diversity optimization algorithm.

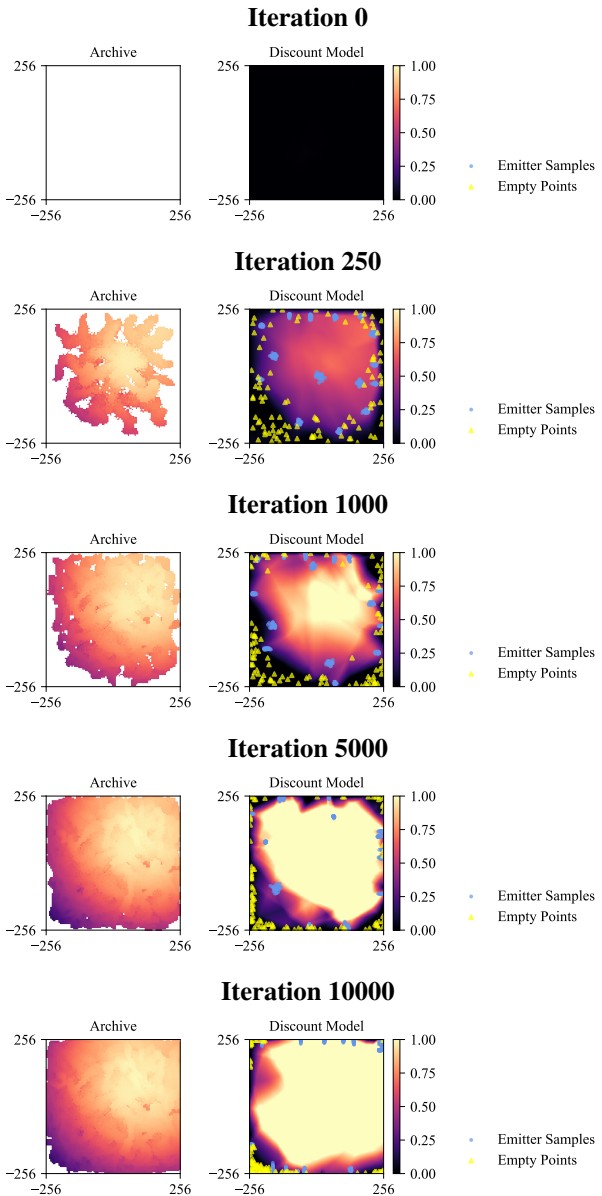

Figure 8: Progression of the archive and discount model in DMS in the 2D LP (Sphere) benchmark. The left heatmap shows the archive, while the right heatmap shows the discount model. To plot the discount model, we computed its output at points in a $200 \times 200$ grid in measure space. The discount model heatmap also shows the dataset $\mathcal{D}_A$ of points on a given iteration — blue circles indicates points created with solutions from the emitters, and yellow triangles indicate empty points. On Iteration 0, the discount model initializes to output $f_{min}$ everywhere. On Iteration 250, as the emitters begin to populate the archive, the discount model begins to output higher values in areas that have been explored. However, unexplored areas still maintain low values (shown as dark colors) due to the empty points in the dataset. On further iterations, the discount model outputs higher and higher values as the emitters populate the archive further, until it outputs high values nearly everywhere on Iteration 10000.

Table 3: Welch's one-way ANOVA results in each domain. All $p$-values are less than 0.001.

| | QD Score | Coverage |
|---|---|---|
| 2D LP (Sphere) | Welch's $F(4, 44.8) = 44362.7$ | Welch's $F(4, 44.2) = 32645.6$ |
| 10D LP (Sphere) | Welch's $F(4, 44.8) = 38532.9$ | Welch's $F(4, 43.7) = 51051.2$ |
| 20D LP (Sphere) | Welch's $F(4, 44.0) = 4887.9$ | Welch's $F(4, 45.3) = 18271.4$ |
| 50D LP (Sphere) | Welch's $F(4, 44.1) = 27718.5$ | Welch's $F(4, 44.4) = 55904.6$ |
| 2D LP (Rastrigin) | Welch's $F(4, 45.0) = 14008.9$ | Welch's $F(4, 44.3) = 11766.9$ |
| 10D LP (Rastrigin) | Welch's $F(4, 45.8) = 19493.4$ | Welch's $F(4, 44.6) = 32616.3$ |
| 2D LP (Flat) | Welch's $F(4, 44.2) = 2121.44$ | Welch's $F(4, 44.2) = 2121.4$ |
| 10D LP (Flat) | Welch's $F(4, 47.1) = 25859.6$ | Welch's $F(4, 47.1) = 25859.6$ |
| Arm Repertoire | Welch's $F(4, 46.5) = 10820.9$ | Welch's $F(4, 38.1) = 15340.2$ |
| TA (MNIST) | Welch's $F(3, 8.4) = 16.5$ | Welch's $F(3, 8.6) = 15.0$ |
| TA (F-MNIST) | Welch's $F(3, 8.8) = 160.6$ | Welch's $F(3, 8.8) = 157.4$ |
| LSI (Hiker) | Welch's $F(3, 7.6) = 18.0$ | Welch's $F(3, 7.2) = 86.0$ |

Table 4: Pairwise comparisons (Games-Howell test) of the QD Score of each algorithm.

**2D LP (Sphere), 10D LP (Sphere), 20D LP (Sphere), 50D LP (Sphere)**

| | 2D LP (Sphere) | | | | | 10D LP (Sphere) | | | | | 20D LP (Sphere) | | | | | 50D LP (Sphere) | | | | |
|---|---|---|---|---|---|---|---|---|---|---|---|---|---|---|---|---|---|---|---|---|
| | DMS | CMA-MAE | DDS | MAP-Elites (line) | MAP-Elites | DMS | CMA-MAE | DDS | MAP-Elites (line) | MAP-Elites | DMS | CMA-MAE | DDS | MAP-Elites (line) | MAP-Elites | DMS | CMA-MAE | DDS | MAP-Elites (line) | MAP-Elites |
| DMS | ∅ | > | > | > | > | ∅ | > | > | < | > | ∅ | > | ∅ | > | > | ∅ | > | > | < | > |
| CMA-MAE | < | ∅ | > | < | > | < | ∅ | < | > | > | < | ∅ | > | < | > | < | ∅ | > | > | > |
| DDS | < | < | ∅ | > | > | < | > | ∅ | > | > | ∅ | < | ∅ | > | > | < | < | ∅ | > | > |
| MAP-Elites (line) | < | > | < | ∅ | > | > | < | < | ∅ | > | < | > | < | ∅ | > | > | < | < | ∅ | > |
| MAP-Elites | < | < | < | < | ∅ | < | < | < | < | ∅ | < | < | < | < | ∅ | < | > | < | < | ∅ |

**2D LP (Rastrigin), 10D LP (Rastrigin), 2D LP (Flat), 10D LP (Flat)**

| | 2D LP (Rastrigin) | | | | | 10D LP (Rastrigin) | | | | | 2D LP (Flat) | | | | | 10D LP (Flat) | | | | |
|---|---|---|---|---|---|---|---|---|---|---|---|---|---|---|---|---|---|---|---|---|
| | DMS | CMA-MAE | DDS | MAP-Elites (line) | MAP-Elites | DMS | CMA-MAE | DDS | MAP-Elites (line) | MAP-Elites | DMS | CMA-MAE | DDS | MAP-Elites (line) | MAP-Elites | DMS | CMA-MAE | DDS | MAP-Elites (line) | MAP-Elites |
| DMS | ∅ | > | > | > | > | ∅ | > | > | > | > | ∅ | > | > | > | > | ∅ | > | > | > | > |
| CMA-MAE | < | ∅ | > | > | > | < | ∅ | > | < | > | < | ∅ | < | > | > | < | ∅ | < | > | > |
| DDS | < | < | ∅ | < | > | > | < | ∅ | > | > | < | > | ∅ | > | > | < | > | ∅ | > | > |
| MAP-Elites (line) | < | < | > | ∅ | > | < | > | < | ∅ | > | < | < | < | ∅ | > | < | < | < | ∅ | > |
| MAP-Elites | < | < | > | < | ∅ | < | < | < | < | ∅ | < | < | < | < | ∅ | < | < | < | < | ∅ |

**Arm Repertoire, TA (MNIST), TA (F-MNIST), LSI (Hiker)**

| | Arm Repertoire | | | | | TA (MNIST) | | | | | TA (F-MNIST) | | | | | LSI (Hiker) | | | | |
|---|---|---|---|---|---|---|---|---|---|---|---|---|---|---|---|---|---|---|---|---|
| | DMS | CMA-MAE | DDS | MAP-Elites (line) | MAP-Elites | DMS | CMA-MAE | DDS | MAP-Elites (line) | MAP-Elites | DMS | CMA-MAE | DDS | MAP-Elites (line) | MAP-Elites | DMS | CMA-MAE | DDS | MAP-Elites (line) | MAP-Elites |
| DMS | ∅ | > | > | > | > | ∅ | — | ∅ | > | > | ∅ | > | ∅ | > | > | ∅ | > | ∅ | — | — |
| CMA-MAE | < | ∅ | > | > | > | — | ∅ | ∅ | > | > | < | ∅ | ∅ | > | > | < | ∅ | ∅ | — | ∅ |
| DDS | < | < | ∅ | < | > | ∅ | ∅ | ∅ | > | > | ∅ | ∅ | ∅ | > | > | ∅ | ∅ | ∅ | ∅ | ∅ |
| MAP-Elites (line) | < | < | > | ∅ | > | < | < | < | ∅ | — | < | < | < | ∅ | > | — | — | ∅ | ∅ | — |
| MAP-Elites | < | < | > | < | ∅ | < | < | < | — | ∅ | < | < | ∅ | < | ∅ | — | — | ∅ | — | ∅ |

Table 5: Pairwise comparisons (Games-Howell test) of the Coverage of each algorithm.

| | 2D LP (Sphere) | | | | | 10D LP (Sphere) | | | | | 20D LP (Sphere) | | | | | 50D LP (Sphere) | | | | |
| --- | --- | --- | --- | --- | --- | --- | --- | --- | --- | --- | --- | --- | --- | --- | --- | --- | --- | --- | --- | --- |
| | DMS | CMA-MAE | DDS | MAP-Elites (line) | MAP-Elites | DMS | CMA-MAE | DDS | MAP-Elites (line) | MAP-Elites | DMS | CMA-MAE | DDS | MAP-Elites (line) | MAP-Elites | DMS | CMA-MAE | DDS | MAP-Elites (line) | MAP-Elites |
| DMS | ∅ | > | > | > | > | ∅ | > | ∅ | > | > | ∅ | > | > | > | > | ∅ | > | > | > | > |
| CMA-MAE | < | ∅ | > | > | > | < | ∅ | < | < | > | < | ∅ | < | < | > | < | ∅ | < | < | > |
| DDS | < | < | ∅ | > | > | ∅ | > | ∅ | > | > | < | > | ∅ | < | > | < | > | ∅ | < | > |
| MAP-Elites (line) | < | < | < | ∅ | > | < | > | < | ∅ | > | < | > | > | ∅ | > | < | > | > | ∅ | > |
| MAP-Elites | < | < | < | < | ∅ | < | < | < | < | ∅ | < | < | < | < | ∅ | < | > | > | > | ∅ |

| | 2D LP (Rastrigin) | | | | | 10D LP (Rastrigin) | | | | | 2D LP (Flat) | | | | | 10D LP (Flat) | | | | |
| --- | --- | --- | --- | --- | --- | --- | --- | --- | --- | --- | --- | --- | --- | --- | --- | --- | --- | --- | --- | --- |
| | DMS | CMA-MAE | DDS | MAP-Elites (line) | MAP-Elites | DMS | CMA-MAE | DDS | MAP-Elites (line) | MAP-Elites | DMS | CMA-MAE | DDS | MAP-Elites (line) | MAP-Elites | DMS | CMA-MAE | DDS | MAP-Elites (line) | MAP-Elites |
| DMS | ∅ | > | > | > | > | ∅ | > | ∅ | > | > | ∅ | > | > | > | > | ∅ | > | > | > | > |
| CMA-MAE | < | ∅ | > | > | > | < | ∅ | < | < | > | < | ∅ | < | < | > | < | ∅ | < | < | > |
| DDS | < | < | ∅ | > | > | ∅ | > | ∅ | > | > | < | > | ∅ | < | > | < | > | ∅ | < | > |
| MAP-Elites (line) | < | < | < | ∅ | > | < | > | < | ∅ | > | < | > | > | ∅ | > | < | > | > | ∅ | > |
| MAP-Elites | < | < | < | < | ∅ | < | < | < | < | ∅ | < | < | < | < | ∅ | < | > | > | > | ∅ |

| | Arm Repertoire | | | | | TA (MNIST) | | | | | TA (F-MNIST) | | | | | LSI (Hiker) | | | | |
| --- | --- | --- | --- | --- | --- | --- | --- | --- | --- | --- | --- | --- | --- | --- | --- | --- | --- | --- | --- | --- |
| | DMS | CMA-MAE | DDS | MAP-Elites (line) | MAP-Elites | DMS | CMA-MAE | DDS | MAP-Elites (line) | MAP-Elites | DMS | CMA-MAE | DDS | MAP-Elites (line) | MAP-Elites | DMS | CMA-MAE | DDS | MAP-Elites (line) | MAP-Elites |
| DMS | ∅ | > | < | < | > | ∅ | − | ∅ | > | > | ∅ | > | ∅ | > | > | ∅ | > | ∅ | − | − |
| CMA-MAE | < | ∅ | < | < | > | − | ∅ | ∅ | > | > | < | ∅ | ∅ | > | > | < | ∅ | ∅ | ∅ | ∅ |
| DDS | > | > | ∅ | > | > | ∅ | ∅ | ∅ | ∅ | ∅ | ∅ | ∅ | ∅ | > | > | ∅ | ∅ | ∅ | − | ∅ |
| MAP-Elites (line) | > | > | < | ∅ | > | < | < | ∅ | ∅ | − | < | < | < | ∅ | > | − | − | ∅ | ∅ | ∅ |
| MAP-Elites | < | < | < | < | ∅ | < | < | ∅ | − | ∅ | < | < | < | < | ∅ | − | ∅ | ∅ | ∅ | ∅ |

## C HYPERPARAMETERS

Table 6: Hyperparameters.

| | 2D LP (Sphere) | 10D LP (Sphere) | 2D LP (Rastrigin) | 10D LP (Rastrigin) | 2D LP (Flat) | 10D LP (Flat) | Arm Repertoire | TA (MNIST) | TA (F-MNIST) | LSI (Hiker) |
|---|---|---|---|---|---|---|---|---|---|---|
| **DMS** | | | | | | | | | | |
| Number of emitters $W$ | 15 | 15 | 15 | 15 | 15 | 15 | 15 | 5 | 5 | 1 |
| Emitter batch size $\lambda$ | 36 | 36 | 36 | 36 | 36 | 36 | 36 | 36 | 36 | 36 |
| Initial step size $\sigma_0$ | 0.5 | 0.5 | 0.5 | 0.5 | 0.5 | 0.5 | 0.2 | 0.1 | 0.1 | 0.02 |
| Archive learning rate $\alpha$ | 0.1 | 0.1 | 0.1 | 0.1 | 0.1 | 0.1 | 0.001 | 0.001 | 0.1 | 1.0 |
| Restart rule | Basic | 100 | Basic | 100 | Basic | 100 | Basic | 50 | 50 | Basic |
| Selection rule | $\mu$ | $\mu$ | $\mu$ | $\mu$ | $\mu$ | $\mu$ | $\mu$ | $\mu$ | $\mu$ | $\mu$ |
| Empty points $n_{empty}$ | 100 | 100 | 100 | 100 | 100 | 100 | 100 | 100 | 100 | 100 |
| Initial points $n_{init}$ | 1000 | 1000 | 1000 | 1000 | 1000 | 1000 | 1000 | 1000 | 1000 | 1000 |
| **CMA-MAE** | | | | | | | | | | |
| Number of emitters $W$ | 15 | 15 | 15 | 15 | 15 | 15 | 15 | 5 | 5 | 1 |
| Emitter batch size $\lambda$ | 36 | 36 | 36 | 36 | 36 | 36 | 36 | 36 | 36 | 36 |
| Initial step size $\sigma_0$ | 0.5 | 0.5 | 0.5 | 0.5 | 0.5 | 0.5 | 0.2 | 0.1 | 0.1 | 0.02 |
| Archive learning rate $\alpha$ | 0.01 | 0.01 | 0.01 | 0.01 | 0.01 | 0.01 | 0.01 | 0.001 | 0.1 | 0.01 |
| Restart rule | Basic | Basic | Basic | Basic | Basic | Basic | Basic | 50 | 50 | Basic |
| Selection rule | $\mu$ | $\mu$ | $\mu$ | $\mu$ | $\mu$ | $\mu$ | $\mu$ | $\mu$ | $\mu$ | $\mu$ |
| **DDS** | | | | | | | | | | |
| Number of emitters $W$ | 15 | 15 | 15 | 15 | 15 | 15 | 15 | — | — | — |
| Emitter batch size $\lambda$ | 36 | 36 | 36 | 36 | 36 | 36 | 36 | — | — | — |
| Initial step size $\sigma_0$ | 1.5 | 1.5 | 1.5 | 1.5 | 1.5 | 1.5 | 0.5 | — | — | — |
| Bandwidth $h$ | 25.6 | 5.12 | 25.6 | 5.12 | 25.6 | 5.12 | 10.0 | — | — | — |
| Buffer size | 10,000 | 10,000 | 10,000 | 10,000 | 10,000 | 10,000 | 10,000 | — | — | — |
| Restart rule | No Imp. | No Imp. | No Imp. | No Imp. | No Imp. | No Imp. | No Imp. | — | — | — |
| Selection rule | Filter | Filter | Filter | Filter | Filter | Filter | Filter | — | — | — |
| **MAP-Elites (line)** | | | | | | | | | | |
| $\lambda$ (batch size) | 540 | 540 | 540 | 540 | 540 | 540 | 540 | 180 | 180 | 36 |
| $\sigma_1$ | 0.5 | 0.5 | 0.5 | 0.5 | 0.5 | 0.5 | 0.5 | 0.1 | 0.1 | 0.1 |
| $\sigma_2$ | 0.2 | 0.2 | 0.2 | 0.2 | 0.2 | 0.2 | 0.2 | 0.2 | 0.2 | 0.2 |
| **MAP-Elites** | | | | | | | | | | |
| $\lambda$ (batch size) | 540 | 540 | 540 | 540 | 540 | 540 | 540 | 180 | 180 | 36 |
| $\sigma$ | 0.5 | 0.5 | 0.5 | 0.5 | 0.5 | 0.5 | 0.5 | 0.1 | 0.1 | 0.1 |

Table 6 lists the hyperparameters of DMS and all baseline algorithms. All algorithms run for 10,000 iterations. The number of solutions generated and evaluated on each iteration is equal across all algorithms. In DMS, CMA-MAE, and DDS, this number is equivalent to the number of emitters $W$ times the emitter batch size $\lambda$. In the two MAP-Elites algorithms, this number is equivalent to the batch size $\lambda$. For DDS, we use the KDE version ("DDS-KDE") (Lee et al., 2024). In all domains, the objective is normalized to be between 0 and 1, so the minimum objective $f_{min}$ is set to 0. For the benchmark domains, parameters for the baselines are adapted from prior work (Fontaine & Nikolaidis, 2023; Lee et al., 2024).

**Archive.** In each domain, all algorithms use the same archive configuration. In benchmark domains, the archive has 10,000 cells, arranged as a $100 \times 100$ grid for domains with 2D measure spaces: 2D LP (Sphere), 2D LP (Rastrigin), 2D LP (Flat), Arm Repertoire. The cells are arranged as a 10,000-cell CVT archive for domains with 10D measure spaces: 10D LP (Sphere), 10D LP (Rastrigin), 10D LP (Flat). In QDDM domains, the archive is a CVT archive consisting of centroids sampled from

the dataset. There are 1,000 cells in the archive for TA (MNIST) and TA (F-MNIST), and 10,000 cells in the archive for LSI (Hiker). The same CVT is used across all trials of all algorithms per domain (as opposed to randomly regenerating the CVT every trial).

**Restart Rule.** The restart rule refers to the conditions upon which emitters are restarted from solutions in the archive. "No Imp." refers to restarting when the emitter no longer discovers solutions that are added to the archive (i.e., solutions that improve the archive) (Fontaine et al., 2020). "Basic" refers to restarting only when default CMA-ES (Hansen, 2016) termination rules are met. An integer value $R$ refers to restarting every $R$ iterations (Fontaine & Nikolaidis, 2023). We study the role of the restart rule in DMS in further detail in Appendix D.

**Discount Model Architecture and Training.** In all domains except for LSI (Hiker), the discount model is a three-layer MLP with layer sizes $[k, 128, 128, 1]$, where $k$ is the dimensionality of the measure space. In LSI (Hiker), the architecture differs slightly in that measures (images) are embedded with the vision transformer of CLIP (Radford et al., 2021) before being passed into a three-layer MLP with layer sizes $[512, 128, 128, 1]$. Beyond that, the details of all MLPs are identical. There is ReLU activation after every layer except the output layer. Inputs to the network are normalized to the range $[-1, 1]$ based on the bounds of the measure space; in the case of images, these bounds are assumed to be $[0, 1]$. Networks are instantiated with the default PyTorch initialization. The discount models are trained with an Adam optimizer with settings of learning rate $\alpha = 0.001$ and $\beta_1 = 0.9$, $\beta_2 = 0.999$. The loss function is mean squared error (MSE), and we train with a batch size of 32. Each iteration (including during the initial training of the model to output $f_{min}$), the model trains until an average cutoff loss of at most 0.05 is reached over the whole dataset $\mathcal{D}_A$, with a maximum of five epochs allowed. In practice, we found that training almost always required only one epoch to reach a cutoff loss of 0.05. The optimizer is maintained throughout the entire run.

## C.1 ON THE CHOICE OF MLP DISCOUNT MODELS

Our selection of an MLP as the discount model (as opposed to kernel-based methods or more complex architectures) is motivated by two reasons. First, MLPs are relatively straightforward to implement and train. In designing DMS, we focused on making the discount model output accurate discount values. For a trainable model like an MLP, this process breaks down into two components: providing the correct training data for the discount values, and ensuring the model can learn to output those values. Prior work on compositional pattern-producing networks (CPPNs) (Ha, 2016) shows that MLPs with similar setups as our discount models can be trained to output complex training data like images, giving us confidence that the MLPs can accurately represent the discount function. Thus, we are able to focus on providing the correct training data (i.e., the targets described in Sec. 5). Second, choosing a neural network architecture like the MLP opens the door to scale to more complex neural network-based models in the future. Namely, now that we know how to train the MLP, we believe it will be feasible to scale to more complex domains by inserting larger architectures like transformers.

## C.2 ON NEAREST NEIGHBOR SEARCH IN ARCHIVES

A key consideration in using CVT archives is how solutions can be assigned to centroids. This process entails a nearest neighbor search in the measure space to identify the closest centroid to each new solution. In lower-dimensional measure spaces, this search can be efficiently performed with a $k$-D tree (Bentley, 1975) in logarithmic time. However, as the dimensionality of the measure space grows, the performance of $k$-D trees degrades to that of brute force. For our experiments, we were able to use $k$-D trees for all the benchmark domains and the TA domains. The benchmark domains had sufficiently low-dimensional measure spaces that the $k$-D tree was able to operate efficiently. Meanwhile, although the TA domains had a 784-dimensional measure space, the archives only had 1000 centroids, making brute force performance acceptable. We note that for higher dimensions (10D or more), we further improved efficiency by allowing the $k$-D tree to use multiple workers (8 workers, specifically).

For LSI (Hiker), nearest neighbor search was slightly more involved since the measures were larger RGB images. However, we note that the CLIP score is computed by first embedding images (or text) into a 512-dimensional latent space, and then computing cosine similarity between the latent vectors. Thus, to make our archives efficient, we precomputed the embeddings for all 10,000 archive

centroids. Then, associating new solutions with centroids in the archive could be accomplished by first embedding the new solutions, and then performing nearest neighbor search with the embeddings of the archive centroids. From this perspective, the nearest neighbor search was only over 10,000 centroids, each of 512 dimensions, rather than each being a large image. We performed this search using the `NearestNeighbors` implementation from scikit-learn (Pedregosa et al., 2011).

We anticipate that for QDDM domains, the computational efficiency of nearest neighbor searches will become a major bottleneck, especially since machine learning datasets often consist of millions of images. In such cases, it may suffice to continue to use brute force search, perhaps implemented on a hardware accelerator like a GPU. However, it may also become necessary to turn to approximate nearest neighbor methods like those implemented in the FAISS library (Douze et al., 2024).

# D  ABLATION STUDY

## D.1  ARCHIVE LEARNING RATE

In DMS, the discount model is trained on a dataset $\mathcal{D}_A$ consisting of entries derived from points sampled by the emitters, and entries created from empty cells in the archive (Sec. 5). The sampled points create targets that reproduce the threshold update rule from CMA-MAE (Eq. 1). As such, this target contains an *archive learning rate* $\alpha$ that controls how quickly the discount model adapts its values. Here, we empirically analyze whether this archive learning rate $\alpha$ induces similar effects as the one in CMA-MAE by varying it from 0.0 to 1.0 in the benchmark domains with 2D and 10D measure spaces.

Fig. 9 displays the results of this ablation. We observe that similar to CMA-MAE, low values of $\alpha$ essentially turn DMS into a single-objective optimization algorithm. This can be seen in the low coverage values for $\alpha = 0.0$, which indicate that DMS is only optimizing the objective and not exploring the measure space. We also observe that QD Score and Coverage typically peak around $\alpha = 0.1$ in the LP benchmarks. Due to low distortion, it is easy to explore the measure space in Arm Repertoire, so there is greater focus on optimization, and thus a lower learning rate of $\alpha = 0.001$ is more helpful.

Figure 9: Mean and standard error of the mean of QD Score and Coverage when varying the archive learning rate $\alpha$ in DMS in the benchmark domains. Highlighted lines indicate results from the main paper in Table 1. Mean over 20 trials.

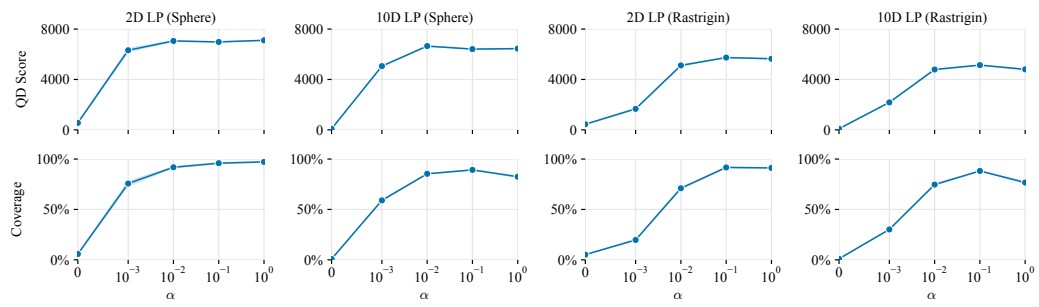

| | 2D LP (Sphere) | | 10D LP (Sphere) | | 2D LP (Rastrigin) | | 10D LP (Rastrigin) | |
|---|---|---|---|---|---|---|---|---|
| | QD Score | Coverage | QD Score | Coverage | QD Score | Coverage | QD Score | Coverage |
| $\alpha = 0.0$ | 556.25 ±38.61 | 5.77 ±0.40% | 82.35 ±1.90 | 0.84 ±0.02% | 443.31 ±18.01 | 5.17 ±0.20% | 89.98 ±3.30 | 1.05 ±0.04% |
| $\alpha = 0.001$ | 6,317.68 ±141.24 | 75.69 ±2.17% | 5,058.38 ±39.31 | 59.00 ±0.48% | 1,666.56 ±42.82 | 19.72 ±0.51% | 2,179.26 ±18.72 | 30.15 ±0.27% |
| $\alpha = 0.01$ | 7,057.29 ±28.30 | 91.83 ±0.61% | **6,649.11 ±21.16** | 85.38 ±0.31% | 5,114.61 ±42.14 | 71.00 ±0.80% | 4,791.17 ±37.63 | 74.67 ±0.64% |
| $\alpha = 0.1$ | 6,978.20 ±17.94 | 95.89 ±0.43% | 6,409.50 ±13.50 | **89.21 ±0.18%** | **5,738.90 ±19.78** | **91.67 ±0.44%** | **5,138.81 ±10.31** | **88.19 ±0.12%** |
| $\alpha = 1.0$ | **7,107.31 ±15.99** | **97.19 ±0.24%** | 6,449.26 ±27.17 | 82.47 ±0.66% | 5,644.14 ±18.91 | 91.20 ±0.30% | 4,802.90 ±37.32 | 76.70 ±0.85% |

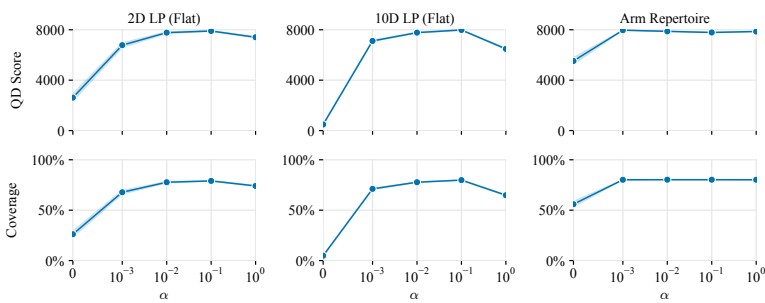

| | 2D LP (Flat) | | 10D LP (Flat) | | Arm Repertoire | |
|---|---|---|---|---|---|---|
| | QD Score | Coverage | QD Score | Coverage | QD Score | Coverage |
| $\alpha = 0.0$ | 2,614.85 ±303.30 | 26.15 ±3.03% | 483.10 ±57.52 | 4.83 ±0.58% | 5,523.36 ±277.61 | 55.94 ±2.77% |
| $\alpha = 0.001$ | 6,777.20 ±187.78 | 67.77 ±1.88% | 7,112.80 ±61.44 | 71.13 ±0.61% | **7,963.44 ±2.47** | **80.15 ±0.01%** |
| $\alpha = 0.01$ | 7,770.30 ±66.72 | 77.70 ±0.67% | 7,772.00 ±42.68 | 77.72 ±0.43% | 7,874.43 ±3.41 | **80.23 ±0.00%** |
| $\alpha = 0.1$ | **7,902.05 ±35.68** | **79.02 ±0.36%** | **7,982.15 ±24.43** | **79.82 ±0.24%** | 7,784.08 ±8.61 | 80.22 ±0.00% |
| $\alpha = 1.0$ | 7,405.05 ±57.10 | 74.05 ±0.57% | 6,483.45 ±68.55 | 64.83 ±0.69% | 7,861.87 ±6.34 | 80.15 ±0.01% |

Figure 10: Mean and standard error of the mean of QD Score and Coverage when varying the number of empty points $n_{empty}$ in DMS in the benchmark domains. Highlighted lines indicate results from the main paper in Table 1. Mean over 20 trials.

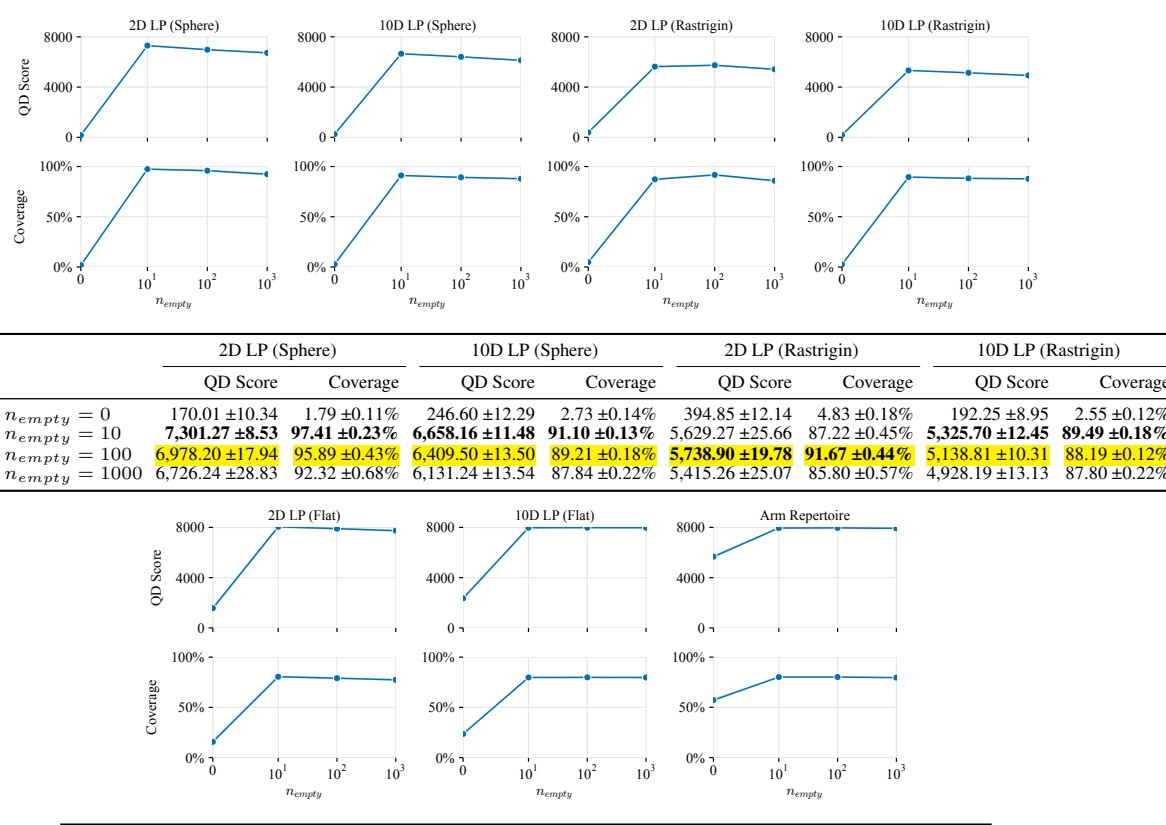

| | 2D LP (Sphere) | | 10D LP (Sphere) | | 2D LP (Rastrigin) | | 10D LP (Rastrigin) | |
|---|---|---|---|---|---|---|---|---|
| | QD Score | Coverage | QD Score | Coverage | QD Score | Coverage | QD Score | Coverage |
| $n_{empty} = 0$ | 170.01 ±10.34 | 1.79 ±0.11% | 246.60 ±12.29 | 2.73 ±0.14% | 394.85 ±12.14 | 4.83 ±0.18% | 192.25 ±8.95 | 2.55 ±0.12% |
| $n_{empty} = 10$ | **7,301.27 ±8.53** | **97.41 ±0.23%** | **6,658.16 ±11.48** | **91.10 ±0.13%** | 5,629.27 ±25.66 | 87.22 ±0.45% | **5,325.70 ±12.45** | **89.49 ±0.18%** |
| $n_{empty} = 100$ | 6,978.20 ±17.94 | 95.89 ±0.43% | 6,409.50 ±13.50 | 89.21 ±0.18% | **5,738.90 ±19.78** | **91.67 ±0.44%** | 5,138.81 ±10.31 | 88.19 ±0.12% |
| $n_{empty} = 1000$ | 6,726.24 ±28.83 | 92.32 ±0.68% | 6,131.24 ±13.54 | 87.84 ±0.22% | 5,415.26 ±25.07 | 85.80 ±0.57% | 4,928.19 ±13.13 | 87.80 ±0.22% |

| | 2D LP (Flat) | | 10D LP (Flat) | | Arm Repertoire | |
|---|---|---|---|---|---|---|
| | QD Score | Coverage | QD Score | Coverage | QD Score | Coverage |
| $n_{empty} = 0$ | 1,573.05 ±97.23 | 15.73 ±0.97% | 2,357.20 ±49.99 | 23.57 ±0.50% | 5,672.67 ±102.41 | 57.24 ±1.01% |
| $n_{empty} = 10$ | **8,049.20 ±63.60** | **80.49 ±0.64%** | 7,975.20 ±25.60 | 79.75 ±0.26% | 7,953.08 ±3.00 | 80.12 ±0.01% |
| $n_{empty} = 100$ | 7,902.05 ±35.68 | 79.02 ±0.36% | **7,982.15 ±24.43** | **79.82 ±0.24%** | **7,963.44 ±2.47** | **80.15 ±0.01%** |
| $n_{empty} = 1000$ | 7,738.75 ±50.29 | 77.39 ±0.50% | 7,972.80 ±32.19 | 79.73 ±0.32% | 7,924.83 ±8.21 | 79.58 ±0.08% |

Table 7: Mean and standard error of the mean of QD Score and Coverage with a basic restart rule and a restart rule of 100 in the benchmark domains. Highlighted lines indicate results from the main paper in Table 1. Mean over 20 trials.

| | 2D LP (Sphere) | | 10D LP (Sphere) | | 2D LP (Rastrigin) | | 10D LP (Rastrigin) | |
|---|---|---|---|---|---|---|---|---|
| | QD Score | Coverage | QD Score | Coverage | QD Score | Coverage | QD Score | Coverage |
| $restart = basic$ | **6,978.20 ±17.94** | **95.89 ±0.43%** | 5,636.33 ±17.84 | 82.58 ±0.23% | **5,738.90 ±19.78** | **91.67 ±0.44%** | 4,298.35 ±22.15 | 76.94 ±0.39% |
| $restart = 100$ | 6,472.23 ±9.94 | 85.76 ±0.08% | **6,409.50 ±13.50** | **89.21 ±0.18%** | 5,304.59 ±4.11 | 84.19 ±0.05% | **5,138.81 ±10.31** | **88.19 ±0.12%** |

| | 2D LP (Flat) | | 10D LP (Flat) | | Arm Repertoire | |
|---|---|---|---|---|---|---|
| | QD Score | Coverage | QD Score | Coverage | QD Score | Coverage |
| $restart = basic$ | 7,902.05 ±35.68 | 79.02 ±0.36% | 2,064.10 ±22.56 | 20.64 ±0.23% | **7,963.44 ±2.47** | **80.15 ±0.01%** |
| $restart = 100$ | **8,390.00 ±21.51** | **83.90 ±0.22%** | **7,982.15 ±24.43** | **79.82 ±0.24%** | 7,659.79 ±4.68 | 77.38 ±0.03% |

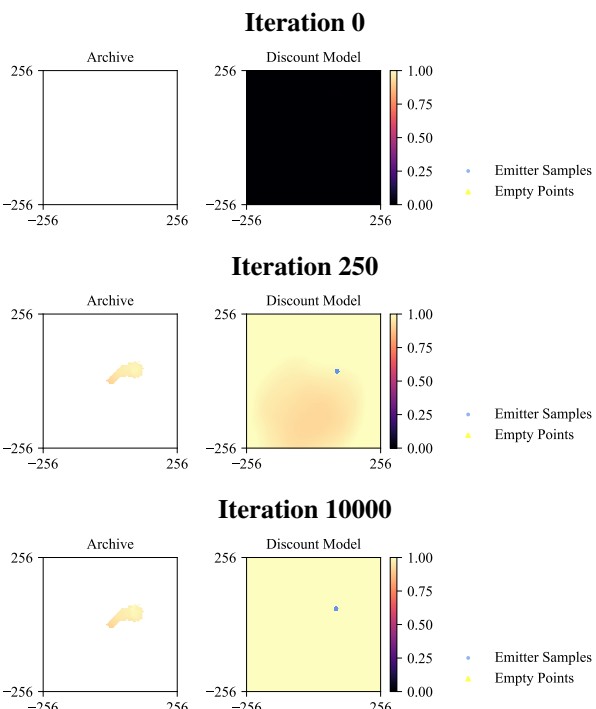

Figure 11: Similar to Fig. 8, this figure shows how the archive and discount model in DMS progress across iterations. However, this time, DMS does not train the discount model with any empty points, i.e., $n_{empty} = 0$. As a result, the discount model takes on arbitrary values in areas of the measure space that have not been explored yet, as evinced by the high values across the discount model heatmap on Iteration 250 and 10000. Because the discount values are high everywhere, the emitters in DMS mistakenly believe they have explored all areas of the measure space, even though the archive is essentially empty.

## D.2 EMPTY POINTS

The second set of entries in the dataset created by DMS are "empty points," which originate as the centers of unoccupied cells in the archive (Sec. 5). To understand the necessity of these points, we vary the number of such points $n_{empty}$ from 0 (no empty points) to 1000 in the benchmark domains with 2D and 10D measure spaces.

Fig. 10 shows the results of this ablation. When there are no empty points, both QD Score and Coverage drop because the discount model takes on arbitrary values in areas of the measure space that have not been explored yet, as shown in Fig. 11. Arbitrary values make it appear as if those areas have already been explored. In contrast, when $n_{empty} > 0$, performance increases because the discount function now outputs the minimum objective $f_{min}$ in those areas, which reflects that those areas have not been explored yet. Surprisingly, across all domains, performance remains relatively even with respect to the number of empty points — DMS with $n_{empty} = 10$, $n_{empty} = 100$, and $n_{empty} = 1000$ all achieve fairly similar scores. However, we note that increasing the number of empty points also increases runtime since the discount model must be trained with these points. In short, these results show that the empty points are a necessary addition to the training set in DMS.

## D.3 RESTART RULE

The "restart rule" in DMS and CMA-MAE refers to the conditions upon which the emitters restart search from a new emitter in the archive (Algorithm 1, line 19-20). By default, emitters in DMS and CMA-MAE use a "basic" restart rule, which makes the emitter restart when the default termination conditions for CMA-ES (Hansen, 2016) are met, e.g., the area of the search distribution becomes too small. While tuning DMS, we found it helpful in some domains to make the emitters restart on a fixed schedule as introduced in prior work (Fontaine & Nikolaidis, 2023), e.g., restarting every 100 iterations. To better understand the effect of the restart rule in DMS, we thus present an ablation where we run DMS in the benchmark domains with both a "basic" restart rule and a restart rule of 100.

Table 7 shows the results of this ablation. We observe that in the 2D LP benchmarks and Arm Repertoire, the "basic" restart rule usually achieves better performance, while in the 10D LP benchmarks, the restart rule of 100 always achieves better performance. The difference is particularly prominent in 10D LP (Flat), where the "basic" restart rule receives a mean QD Score of 2,064.10 while the restart rule of 100 receives a mean QD Score of 7,982.15. As such, it seems the restart rule of 100 is particularly helpful in the benchmarks with higher-dimensional measure spaces.

We speculate that this result occurs due to "hacking" of the discount model. In short, it may be possible for an emitter to generate solutions that achieve similar measures. The discount model then updates to reflect that that specific area of measure space now has a high discount value. However, the emitter can now slightly modify its distribution such that it generates solutions in a nearby area that still has a low discount value. It is especially easy to do this in a high-dimensional measure space because there are exponentially more "nearby areas" with low discount values. Thus, with a basic restart rule, the emitter can continue this process and receive high discount values. On the other hand, a fixed restart rule forces the emitter to restart before it can converge to a small distribution that hacks the discount model.

## E DOMAIN DETAILS

### E.1 LINEAR PROJECTION

Introduced in prior work (Fontaine et al., 2020) and extended in later work (Lee et al., 2024), LP considers a measure function $\boldsymbol{m}$ that maps an $n$-dimensional solution space to a $k$-dimensional measure space. Given $r = \frac{n}{k}$, the measure values are bounded by $[-5.12 \cdot r, 5.12 \cdot r]^k$ and defined as:

$$m(\boldsymbol{\theta}) = \left( \sum_{i=jr+1}^{(j+1)r} \text{clip}(\theta_i) : j \in \{0, ..., k-1\} \right)$$

$$\text{clip}(\theta_i) = \begin{cases} \theta_i & \text{if } |\theta_i| \leq 5.12 \\ 5.12/\theta_i & \text{otherwise} \end{cases}$$

where $\theta_i$ is the $i$th component of $\boldsymbol{\theta}$ (one-indexed). The measure function, $m(\boldsymbol{\theta})$, partitions $\boldsymbol{\theta}$ into $k$ contiguous, non-overlapping blocks of size $r$, applies clip($\cdot$) element-wise, and then sums each block, thus producing a $k$-dimensional measure vector. The clip($\cdot$) function bounds $\theta_i$ to the interval $[-5.12, 5.12]$, so $m(\boldsymbol{\theta})$ is bound by $[-5.12 \cdot r, 5.12 \cdot r]^k$.

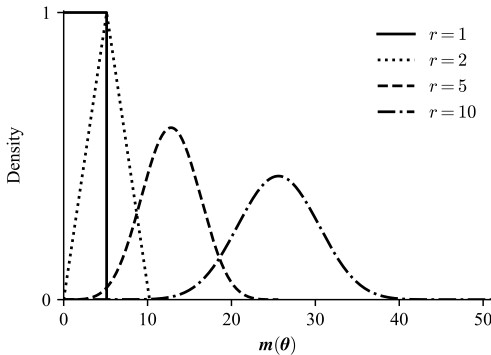

Figure 12: Irwin-Hall distribution when summing $r$ uniformly random variables with range $[-5.12, 5.12]$.

To understand how the measure function creates *distortion*, consider if we randomly sample each solution value $\theta_i$ from the range $[-5.12, 5.12]$. Since each measure value is made by summing a block of $r$ values from the solution vector, the distribution of each measure value forms an Irwin-Hall distribution (Fig. 12). Importantly, observe that for higher values of $r$, the probability of attaining extreme measure values becomes less and less likely. For example, with $r = 2$, if we sample all $\theta_i$ uniformly at random, it is quite likely that we obtain a solution with measures close to 0 or 10. In comparison, for $r = 10$, there is virtually no probability of obtaining the extreme values in the range $[0, 10]$ or $[40, 50]$. In other words, there is higher distortion because a larger portion of solution space maps to the relatively small region of measure space in the center of the distribution. However, it is possible to overcome this distortion and reach the edges of the measure space by intelligently sampling $\theta_i$.

Our experiments instantiate the LP domain with the Sphere and Rastrigin objectives from the blackbox optimization benchmark (Hansen et al., 2010):

$$f_{\text{Sphere}}(\boldsymbol{\theta}) = \sum_{i=1}^{n} \theta_i^2$$

$$f_{\text{Rastrigin}}(\boldsymbol{\theta}) = 10n + \sum_{i=1}^{n} [\theta_i^2 - 10\cos(2\pi\theta_i^2)]$$

While these objectives require minimization by default, we convert them for maximization by taking their negative values. Furthermore, following prior work (Fontaine et al., 2020), we shift the global optimum to $\theta_i = 5.12 * 0.4 = 2.048$. We also normalize the objective to the range $[0, 1]$ — to do this, we consider the minimum possible value of the objective to be at $\theta_i = -5.12 - 0.4 * 5.12 = -7.168$.

In addition to Sphere and Rastrigin, we consider the Flat objective from prior work (Lee et al., 2024):

$$f_{\text{Flat}}(\boldsymbol{\theta}) = 1$$

The Flat objective turns the domain into a diversity optimization domain — since all solutions have the same objective value, an algorithm only needs to find solutions with diverse measures.

For each objective, we considered solution dimensionality $n = 100$ and measure space dimensionality $k = \{2, 10\}$. Thus, we obtained six LP benchmarks, named according to their measure space dimensionality and objective function: **2D LP (Sphere)**, **10D LP (Sphere)**, **2D LP (Rastrigin)**, **10D LP (Rastrigin)**, **2D LP (Flat)**, **10D LP (Flat)**.

### E.2 ARM REPERTOIRE

Arm Repertoire (Cully & Demiris, 2017; Vassiliades & Mouret, 2018) considers solutions $\boldsymbol{\theta} \in \mathbb{R}^n$ that represent the $n$ joint angles of a planar robotic arm with $n$ joints and $n$ links of length 1. The measure function, $\boldsymbol{m}(\boldsymbol{\theta})$, uses forward kinematics to compute the $(x, y)$ position of the end effector — thus, the measure space is two-dimensional. The objective function seeks to minimize the variance between each joint angle:

$$f(\boldsymbol{\theta}) = -\text{var}(\boldsymbol{\theta})$$

Our experiments consider $n = 100$. We also add 1 to the objective to normalize it to a maximum value of 1.

### E.3 TRIANGLE ARRANGEMENT (TA)

This QDDM domain considers arranging triangles to create images, as inspired by prior work (Tian & Ha, 2022). We consider grayscale images in our work, with the triangles all drawn on a black background. Each triangle is parameterized by the following eight values:

$$(x_0, y_0, x_1, y_1, x_2, y_2, brightness, a)$$

$(x_0, y_0)$, $(x_1, y_1)$, and $(x_2, y_2)$ are the vertices of the triangle, and the triangle is shaded/filled according to the given $brightness$ and alpha (transparency) value $a$. A single solution vector $\boldsymbol{\theta}$ consists of the parameters for 30 triangles concatenated together; thus, each solution has $30 * 8 = 240$ parameters. The triangles are drawn in the order specified in the solution vector, i.e., the last triangle would end up on top of the first triangle if they intersect. To render the triangles as a raster image, we adapt the JAX (Bradbury et al., 2018) implementation from EvoJAX (Tang et al., 2022; EvoJAX, 2022).

We define the measure space as the space of $28 \times 28$ grayscale images, making it 784-dimensional. Each dimension has bounds $[0, 1]$, i.e., the possible brightness values of a grayscale pixel. For each solution $\boldsymbol{\theta}$, we compute the measures by rendering a $28 \times 28$ raster image of its triangles. We specify desired measures by sampling 1000 images from either MNIST (Lecun et al., 1998) or Fashion MNIST (Xiao et al., 2017), leading to the two versions of this domain considered in this work, **TA (MNIST)** and **TA (F-MNIST)**. These 1000 images form the centroids of the CVT archive. To determine the closest centroid when inserting solutions into the CVT archive, we compute the Euclidean distance between the solution's measures (i.e., its rendered image) and each centroid.

For each solution, the objective is the negative mean squared error between the solution's measures and the centroid to which it is closest. To normalize this objective to a range of $[0, 1]$, we add 1 to it. This objective enables solutions that resemble an MNIST image to replace solutions that do not.

### E.4 LATENT SPACE ILLUMINATION (LSI)

We construct this QDDM domain by adapting prior work that introduced LSI domains with 2D measures (Fontaine & Nikolaidis, 2021; 2023), and prior work that shows how to guide StyleGAN3 to generate images that match text prompts (Herrera-Berg, 2021). In our domain, which we refer to

as **LSI (Hiker)**, the goal is to generate face images of hikers in different landscapes, e.g., a hiker who is ready for the mountains.

To that end, we consider StyleGAN3 (Karras et al., 2021a) as our generative model, specifically the pretrained `stylegan3-t-ffhqu-256x256.pkl` that generates $256 \times 256$ RGB images of faces. Each solution $\boldsymbol{\theta}$ is a latent vector in the $w$-space of StyleGAN3, making it 512-dimensional. Note that $w$-space differs from $w^+$-space, which assigns a distinct style vector to each layer of the GAN and is much higher-dimensional (e.g., 8,192- or 7,168-dimensional).

We consider the space of $256 \times 256 \times 3$ RGB images as the measure space. The measure function $\boldsymbol{m}(\boldsymbol{\theta})$ outputs the face image generated by passing the solution $\boldsymbol{\theta}$ through StyleGAN3. To specify desired measures, we sample 10,000 landscape images from the LHQ256 (Skorokhodov et al., 2021) dataset. These images form the centroids of a CVT archive, and when inserting solutions, we compute the closest centroid with CLIP score (Radford et al., 2021), specifically with the `ViT-B/32` model of CLIP. The intuition is that the CLIP score will cause a hiker to be associated with the centroid/landscape that is semantically closest to them. For example, a hiker wearing a thick jacket is most suited for a cold landscape like the mountains, while a hiker wearing thin clothes is most suited for the beach (Fig. 2).

The objective in this domain considers several factors:

1. $f_{prompt}$: CLIP score between the generated image and the prompt "A photo of the face of a hiker."

2. $f_{centroid}$: CLIP score between the generated image and the centroid to which the solution is assigned. This results in images that more closely match the landscape specified in the centroid.

3. $f_{reg}$: We sample 10,000 points in $w$-space and compute their mean and standard deviation. If the latent vector $\boldsymbol{\theta}$ strays too far outside this distribution, we apply an L2 penalty.

The final objective is computed as:

$$f = \frac{f_{prompt} + f_{centroid}}{2} - f_{reg}$$

If an algorithm stays within the training distribution of StyleGAN3, it should not incur any regularization penalty, and the objective should stay between 0 and 1. However, the objective can become negative if an algorithm goes outside the training distribution and thus receives high values for $f_{reg}$.

## F    IMPLEMENTATION

**Compute Resources.** We run our experiments on a workstation with a 64-core (128-thread) AMD Ryzen Threadripper, NVIDIA RTX A6000 GPU, and 64GB of RAM.

**Compute Usage.** In benchmark domains, we run all 20 trials of each algorithm in parallel. In QDDM domains, we run algorithms serially. For wallclock times, refer to Table 2. In all domains, we train the discount model of DMS on the GPU. The discount model typically requires 300-400MB of GPU memory. In TA (MNIST) and TA (F-MNIST), we accelerate rendering of the triangles on the GPU. In LSI (Hiker), we accelerate StyleGAN3 and CLIP with the GPU. The final experimental results required 100GB of storage.

**Preliminary Experiments.** We estimate that developing DMS and tuning the baselines required about the same amount of compute as was used in the final experiments and ablations, viz., all our preliminary results occupied another 100GB of storage.

**Software.** We implement DMS and all baselines with the pyribs (Tjanaka et al., 2023) library, which is available under the MIT License.

**Datasets.** We use the MNIST (Lecun et al., 1998), Fashion MNIST (Xiao et al., 2017), and LHQ (Skorokhodov et al., 2021) datasets in our work. MNIST is in the public domain, Fashion MNIST is under an MIT License, and LHQ is available under the CC BY 2.0 license.

## G    Differentiable Quality Diversity in QDDM Domains

While our work considers a black-box QD setting as defined in Sec. 2, prior work (Fontaine & Nikolaidis, 2021) introduced differentiable quality diversity (DQD), a QD setting where the objective and measure functions are first-order differentiable. Since DQD methods make assumptions that DMS and our other baselines do not, viz., that the objective and measures are differentiable, we have not included DQD methods in our experiments. Nevertheless, here we discuss considerations in applying DQD methods to QDDM domains.

The primary consideration is that DQD methods, such as CMA-MEGA (Fontaine & Nikolaidis, 2021), were designed with the assumption of a high-dimensional solution space that could be reduced to a low-dimensional objective-measure space, under the assumption that the measure space is low-dimensional. To elaborate, during operation, CMA-MEGA and other DQD methods compute a gradient for the objective and for each measure. In low-dimensional measure spaces, e.g., a 2D measure space, this works well because only a few gradients, 1 for the objective and 2 for the measures in this case, need to be computed. In contrast, in QDDM domains, the measure spaces are much higher-dimensional than even the solution space. In LSI (Hiker), the solution space is 512D while the measure space is 196,608D ($256 \times 256 \times 3$ RGB image). In the TA domains, the solution space is 240D while the measure space is 784D (it is also worth noting that the TA domains are non-differentiable due to the rendering process). As such, running DQD in these domains would mean computing up to 196,608 measure gradients on every iteration, which is computationally intractable. Given this limitation, we believe developing a DQD method that operates in QDDM domains is an exciting avenue for future work.

