# OpenReview forum: "Discount Model Search for Quality Diversity Optimization in High-Dimensional Measure Spaces"
_ICLR.cc/2026/Conference — ICLR 2026 Oral_

### Official Review · Reviewer_TnHM · 2025-10-19

**Soundness:** 2
**Presentation:** 2
**Contribution:** 2
**Rating:** 4
**Confidence:** 3

**Summary:**

This paper proposes Discount Model Search, a Quality-Diversity algorithm to address stagnation in high-dimensional measure spaces caused by discrete histograms in state-of-the-art methods like CMA-MAE. DMS uses a smooth neural network-based discount model to distinguish solutions with similar measures, guiding exploration effectively. It also introduces the Quality Diversity with Datasets of Measures setting, where users specify desired measures via datasets instead of hand-designing measure functions. DMS is evaluated on 7 benchmark domains and 3 QDDM domains , outperforming baselines like CMA-MAE, DDS, and MAP-Elites in most cases.

**Strengths:**

1. By replacing CMA-MAE’s discrete histogram with a continuous, learnable discount model, DMS addresses the limitation of identical discount values for similar measures in high-dimensional spaces.
2. The QDDM framework eliminates the need for tedious manual measure function design by leveraging datasets. This expands QD’s applicability to vision and creative domains.
3. Experiments cover both standard QD benchmarks (testing distortion and scalability) and novel QDDM tasks (testing real-world applicability). Ablation studies rigorously verify key components (e.g., empty points prevent arbitrary discount values, archive learning rate α balances quality/diversity).

**Weaknesses:**

1. The paper only mentions Kent et al. (2022)’s continuous QD Score as an “evaluation tool” but fails to deeply contrast DMS’s core innovation with Kent et al.’s static metric.\
2. DMS uses MLPs for the discount model for non-image measures) but provides no justification for why MLPs outperform other models (e.g., kernel methods, transformers for image measures). For high-dimensional image measures in LSI, the MLP’s ability to capture complex semantic relationships is unproven.
3. Uses Euclidean distance and LSI uses CLIP scores for centroid matching, but the paper does not compare these choices with alternatives (e.g., cosine similarity for images) or validate their impact on performance.
4. The LP benchmark only tests up to 16-dimensional measure spaces.
5. DMS adds discount model training, but the paper does not quantify its computational cost (e.g., runtime, GPU memory) relative to CMA-MAE.

**Questions:**

1. Could you provide a detailed comparison between DMS’s learnable continuous discount model and Kent et al. (2022)’s continuous QD Score, explaining how DMS’s model-guided exploration advances beyond the static metric in Kent et al.’s work, especially in addressing high-dimensional measure space distortion?
2. For the discount model architecture (e.g., MLPs with [k, 128, 128, 1] layers), could you provide justification for why MLPs are superior to alternative models (e.g., kernel methods for low-dimensional measures, transformers for high-dimensional image measures)?
3. In the LSI (Hiker) domain, DMS only achieves 3.77% archive coverage. Could you analyze the cause of this low coverage? Additionally, propose and test potential strategies to improve coverage.
4. The authors attribute DMS’s similar performance to CMA-MAE in TA (MNIST) to “discount model noise.” Could you quantify this noise (e.g., mean squared error between predicted and ideal discount values) and explicitly analyze how it disrupts improvement rankings and emitter updates? Also, explain why this noise has a more pronounced impact in TA (MNIST) than in other domains.
5. Could you quantify the computational overhead of DMS’s discount model training (e.g., per-iteration runtime, GPU memory usage) relative to CMA-MAE across all domains (e.g., 10D LP, LSI)? For large QDDM datasets (e.g., 10,000 LHQ images), also clarify the measures taken to ensure DMS remains computationally efficient.

---

> ### Author Response · Authors · 2025-11-23
> **Author Response (Part 1)**
>
> Thank you for your detailed feedback on our work. Below we respond to your comments and questions. We have grouped together related items, and we have revised our paper where appropriate.
>
> > W1. The paper only mentions Kent et al. (2022)’s continuous QD Score as an “evaluation tool” but fails to deeply contrast DMS’s core innovation with Kent et al.’s static metric.
> > Q1. Could you provide a detailed comparison between DMS’s learnable continuous discount model and Kent et al. (2022)’s continuous QD Score, explaining how DMS’s model-guided exploration advances beyond the static metric in Kent et al.’s work, especially in addressing high-dimensional measure space distortion?
>
> We wish to clarify that we do not reference Kent et al.’s continuous QD Score in our work. However, we do reference BOP-Elites (Kent et al., 2025) in our related work as an example of a work that integrates models with QD algorithms. Specifically, BOP-Elites leverages surrogate models to avoid expensive solution evaluations — in contrast, DMS uses a discount model to guide the search process for CMA-ES emitters.
>
> Regarding the difference between DMS and the continuous QD Score, DMS is a QD _algorithm_ that introduces a discount model in order to enhance its search process in high-dimensional measure spaces. In contrast, continuous QD Score is a _metric_ that can be used for evaluating the performance of any QD algorithm, including DMS. Specifically, continuous QD Score was developed as a metric that captures the trade-off between performance on the objective and accuracy in the measure space.
>
> In our work, we use the QD Score to measure performance since all our algorithms output a MAP-Elites-style archive. This is a good setting for using the QD Score since the QD Score sums objective values over all the cells in an archive. Continuous QD Score would be more applicable if we were comparing QD algorithms from both the MAP-Elites and Novelty Search families of algorithms.
>
> Finally, we are unaware of a relationship between the continuous QD Score and high-dimensional measure space distortion. Our understanding is that the continuous QD Score was designed to evaluate QD algorithms without having to discretize the measure space; this concept would apply in both low- and high-dimensional measure spaces. In short, DMS is an algorithm designed to address distortion in high-dimensional measure spaces, while the continuous QD Score is a metric that can be applied to measure any QD algorithm’s performance.

---

> > ### Author Response · Authors · 2025-11-23
> > **Author Response (Part 2)**
> >
> > > W2. DMS uses MLPs for the discount model for non-image measures) but provides no justification for why MLPs outperform other models (e.g., kernel methods, transformers for image measures). For high-dimensional image measures in LSI, the MLP’s ability to capture complex semantic relationships is unproven.
> > > Q2. For the discount model architecture (e.g., MLPs with [k, 128, 128, 1] layers), could you provide justification for why MLPs are superior to alternative models (e.g., kernel methods for low-dimensional measures, transformers for high-dimensional image measures)?
> >
> > Our selection of an MLP as the discount model is motivated by two reasons. First, MLPs are relatively straightforward to implement and train. In designing DMS, we focused on making the discount model output accurate discount values. For a trainable model like an MLP, this process breaks down into two components: providing the correct training data for the discount values, and ensuring the model can learn to output those values. Prior work on compositional pattern-producing networks (CPPNs) (https://blog.otoro.net/2016/04/01/generating-large-images-from-latent-vectors/) shows that MLPs with similar setups as our discount models can be trained to output complex training data like images, giving us confidence that the MLPs can accurately represent the discount function. Thus, we are able to focus on providing the correct training data (i.e., the targets described in Section 5), without having to worry about whether the MLP can learn to match that training data.
> >
> > The second reason is that choosing a neural network architecture like the MLP makes it easy to scale to more complex models in the future. Namely, now that we know how to train the MLP, we believe it will be feasible to scale to more complex domains by replacing the architecture with something larger like a transformer. While it is true that kernel methods could suffice for low-dimensional measures, we do not believe they would scale to the more complex data that we anticipate for QDDM domains.
> >
> > Finally, we wish to clarify our choice of discount model for the image domains. For MNIST, it is true that we use an MLP as the discount model. We felt that this choice was sufficient since MLPs are frequently used for MNIST tasks ([example](https://www.codegenes.net/blog/mnist-mlp-pytorch-high-accuracy/)). For the LSI (Hiker) domain, we agree with the reviewer that MLPs cannot capture complex semantic relationships in images, as the images there are much larger (256x256x3 RGB images). As such, we clarify that we do not use an MLP directly in the LSI domain. Instead, we embed the images with the pretrained vision transformer from CLIP, and then feed the embeddings into an MLP head. The vision transformer remains frozen, while the MLP head is trained.
> >
> > We have added a new discussion on our choice of MLP to Appendix C.1.

---

> > > ### Author Response · Authors · 2025-11-23
> > > **Author Response (Part 3)**
> > >
> > > > W3. Uses Euclidean distance and LSI uses CLIP scores for centroid matching, but the paper does not compare these choices with alternatives (e.g., cosine similarity for images) or validate their impact on performance.
> > >
> > > In our paper, we focus on leveraging discount models to address distortion in high-dimensional measure spaces. Hence, when designing our domains, we sought to make reasonable choices based on prior work. To elaborate, for the benchmark domains (LP and Arm), we drew from prior work [1, 2] that used Euclidean distance as the default for CVT archives. For the TA domains, prior work [3] showed that Euclidean distance was a suitable loss function when optimizing triangle arrangements to match a given image, so we also used Euclidean distance for centroid matching in TA. Finally, for the LSI (Hiker) domain, directly using Euclidean distance seemed unreasonable since we sought to match hiker faces with landscapes. As such, we needed a distance metric that has semantic meaning, which made the CLIP score a reasonable choice. We note that the CLIP score is computed by first embedding the images into a latent space with the vision transformer of CLIP, and then computing cosine similarity between these latent vectors. In the future, we agree that it would be helpful to assess alternative distance metrics, but given our focus on DMS, we believe that such an assessment is out of scope for this paper.
> > >
> > > > W4. The LP benchmark only tests up to 16-dimensional measure spaces.
> > >
> > > To extend our results on the LP benchmark, we ran experiments on 20D and 50D versions of the LP benchmark. We present the results below:
> > >
> > > **20D LP (Sphere)**
> > >
> > > |                   | QD Score             | Coverage           |
> > > | :---------------- | :------------------- | :----------------- |
> > > | DMS               | **7,406.01 ± 23.37** | **95.97 ± 0.07\%** |
> > > | CMA-MAE           | 881.76 ± 72.92       | 9.13 ± 0.80\%      |
> > > | MAP-Elites (line) | 5,936.66 ± 6.90      | 65.86 ± 0.10\%     |
> > > | MAP-Elites        | 3,280.34 ± 94.26     | 35.62 ± 1.10\%     |
> > >
> > > **50D LP (Sphere)**
> > >
> > > |                   | QD Score            | Coverage           |
> > > | :---------------- | :------------------ | :----------------- |
> > > | DMS               | **6,991.00 ± 4.48** | **87.00 ± 0.04\%** |
> > > | CMA-MAE           | 2,327.11 ± 18.78    | 24.21 ± 0.20\%     |
> > > | MAP-Elites (line) | 5,204.80 ± 11.47    | 55.77 ± 0.14\%     |
> > > | MAP-Elites        | 3,870.48 ± 12.47    | 40.89 ± 0.14\%     |
> > >
> > > In both cases, we find that DMS achieves the best performance in terms of both QD Score and Coverage. Note that due to time constraints, we were unable to run DDS, which is computationally expensive, but we will do so for the final version of our paper. We have added these experiments and results to our main paper, including Table 1 and Section 7, and we have added these results to the full results in Appendix B.

---

> > > > ### Author Response · Authors · 2025-11-23
> > > > **Author Response (Part 4)**
> > > >
> > > > > W5. DMS adds discount model training, but the paper does not quantify its computational cost (e.g., runtime, GPU memory) relative to CMA-MAE.
> > > > > Q5. Could you quantify the computational overhead of DMS’s discount model training (e.g., per-iteration runtime, GPU memory usage) relative to CMA-MAE across all domains (e.g., 10D LP, LSI)? For large QDDM datasets (e.g., 10,000 LHQ images), also clarify the measures taken to ensure DMS remains computationally efficient.
> > > >
> > > > Below we provide the wallclock time from running all algorithms in all domains. We see that in the LP domains, the discount model training increases the wallclock time compared to CMA-MAE and other algorithms. However, in the QDDM domains, the difference with CMA-MAE is minimal, as the evaluation of solutions is the bottleneck rather than the algorithm itself. We also note that MAP-Elites and MAP-Elites (line) are the fastest algorithms because they involve the fastest operations. Namely, while CMA-MAE and DMS use CMA-ES emitters, which require manipulating a multivariate Gaussian distribution, MAP-Elites and MAP-Elites (line) involve mutations that are based on fixed Gaussian noise.
> > > >
> > > > |                   | 2D LP (Sphere)   | 10D LP (Sphere)   | 2D LP (Rastrigin) | 10D LP (Rastrigin) | 2D LP (Flat)     | 10D LP (Flat)     | Arm Repertoire   | TA (MNIST)        | TA (F-MNIST)      | LSI (Hiker)         |
> > > > | :---------------- | :--------------- | :---------------- | :---------------- | :----------------- | :--------------- | :---------------- | :--------------- | :---------------- | :---------------- | :------------------ |
> > > > | DMS               | 397.83 ± 0.37    | 876.33 ± 44.22    | 707.20 ± 21.31    | 711.08 ± 30.76     | 683.53 ± 27.73   | 723.34 ± 32.31    | 687.85 ± 27.18   | 489.90 ± 12.55    | 546.60 ± 21.46    | 4,740.24 ± 482.74   |
> > > > | CMA-MAE           | 121.13 ± 0.10    | 333.92 ± 1.70     | 182.07 ± 0.36     | 465.80 ± 3.81      | 165.33 ± 0.25    | 800.12 ± 0.92     | 130.83 ± 0.26    | 495.95 ± 13.12    | 525.42 ± 19.22    | 4,690.12 ± 654.47   |
> > > > | DDS               | 1,918.68 ± 22.42 | 3,535.78 ± 1.53   | 1,957.29 ± 0.87   | 3,598.86 ± 2.56    | 1,937.41 ± 1.51  | 3,581.79 ± 2.05   | 1,577.54 ± 1.70  | --                | --                | --                  |
> > > > | MAP-Elites (line) | **31.10 ± 0.10** | 231.50 ± 0.73     | **39.74 ± 0.14**  | **274.86 ± 0.62**  | **30.84 ± 0.09** | **339.73 ± 0.42** | 39.68 ± 0.10     | **247.21 ± 4.07** | **231.91 ± 0.38** | 3,470.08 ± 12.88    |
> > > > | MAP-Elites        | 39.01 ± 0.21     | **190.25 ± 2.46** | 51.27 ± 0.17      | 367.56 ± 1.57      | 38.41 ± 0.14     | 451.46 ± 0.41     | **32.80 ± 0.08** | 291.98 ± 1.46     | 309.01 ± 3.92     | **3,460.71 ± 5.86** |
> > > >
> > > > In our main paper, we have added Table 2 and Section 7.3 to display and analyze these results.
> > > >
> > > > Regarding GPU memory, DMS typically requires 300-400MB, since the MLP is fairly small. In the LSI (Hiker) domain, we do not count the CLIP model since CMA-MAE also requires CLIP for evaluating solutions. We have added a line describing this usage to Appendix F. CMA-MAE runs entirely on CPU and only uses the GPU for evaluations.
> > > >
> > > > For large QDDM datasets, a variety of methods are available for ensuring DMS remains computationally efficient. In the case of LSI (Hiker), we realized that it would be computationally inefficient to directly compute the CLIP score between images. Note that the CLIP score consists of first embedding the images into a 512D latent space with the vision transformer of CLIP, and then computing the cosine similarity between the latent vectors for the images. Thus, we precomputed the image embeddings for all the centroids in the archive. Then, associating new solutions with centroids in the archive could be accomplished by first embedding the new solutions, and then performing nearest neighbor search with the embeddings of the archive centroids. For the nearest neighbor search, we used the [NearestNeighbors](https://scikit-learn.org/stable/modules/generated/sklearn.neighbors.NearestNeighbors.html) implementation in scikit-learn, which proved fast enough on CPU since there were only 10,000 centroids that were each 512D. For slightly larger datasets, we anticipate that brute force implementations accelerated on the GPU would suffice, and for extremely large datasets, we anticipate that approximate nearest neighbor approaches like those implemented in FAISS (https://github.com/facebookresearch/faiss) would be appropriate. We have added a discussion of these considerations in Appendix C.2.

---

> > > > > ### Author Response · Authors · 2025-11-23
> > > > > **Author Response (Part 5)**
> > > > >
> > > > > > Q3. In the LSI (Hiker) domain, DMS only achieves 3.77% archive coverage. Could you analyze the cause of this low coverage? Additionally, propose and test potential strategies to improve coverage.
> > > > >
> > > > > In general, we intend for LSI (Hiker) to be a challenging domain that illustrates the potential of the QDDM setting. To succeed in this domain, an algorithm must be able to navigate the latent space of StyleGAN as well as the extremely high-dimensional measure space. As such, we believe that DMS’s performance in this domain merely shows that there is still room for improvement in searching in high-dimensional measure spaces. Furthermore, while we acknowledge that the coverage is low as a percentage, we note that it still corresponds to discovering 377 hiker images, which we believe is sufficient for a number of applications.
> > > > >
> > > > > To improve DMS in the future, several approaches are available. One would be to replace the CMA-ES emitters with other algorithms. For example, we could use scalable variants of CMA-ES [4] or Natural Evolution Strategies [5]. These algorithms may be better able to navigate StyleGAN’s latent space, which is 512-dimensional (fairly large for QD algorithms). Another approach, similar to reward shaping in reinforcement learning, is to take a closer look at the domain itself, and determine whether any aspects like the objective are hindering the search.
> > > > >
> > > > > > Q4. The authors attribute DMS’s similar performance to CMA-MAE in TA (MNIST) to “discount model noise.” Could you quantify this noise (e.g., mean squared error between predicted and ideal discount values) and explicitly analyze how it disrupts improvement rankings and emitter updates? Also, explain why this noise has a more pronounced impact in TA (MNIST) than in other domains.
> > > > >
> > > > > We wish to clarify that we propose the idea of discount model noise as a hypothesis to explain the performance of DMS in TA (MNIST), rather than as a definitive explanation. Analyzing the noise, assuming it exists, is non-trivial for several reasons. First, it is difficult to determine ideal discount values. An ideal discount model would be one that seamlessly guides the CMA-ES emitters to discover the optimal archive for a QD problem. However, the discount model and the emitters are quite intertwined — the discount model’s updates depend on solutions output by the emitters, and the updates of the emitter’s distributions depend on the discount model’s outputs. Furthermore, the update rules of CMA-ES are quite complex, as they involve numerous components. Finally, it is unclear what the optimal archive for our benchmarks looks like — we would need to define a new domain where the optimal archive is known. For these reasons, we believe that properly studying this limitation is beyond the scope of this paper, which seeks to address the distortion problem by proposing discount models and developing an algorithm to leverage them. In the text, we have added text in Section 7.1 to clarify that this idea is a hypothesis that would be suitable for future work.
> > > > >
> > > > > [1] D. H. Lee, A. V. Palaparthi, M. C. Fontaine, B. Tjanaka, S. Nikolaidis. “Density Descent for Diversity Optimization.” GECCO 2024.
> > > > >
> > > > > [2] V. Vassiliades, K. Chatzilygeroudis, and J.-B. Mouret. “Using centroidal
> > > > > voronoi tessellations to scale up the multidimensional archive of phenotypic elites algorithm.” IEEE Transactions on Evolutionary Computation, 2017.
> > > > >
> > > > > [3] Y. Tian and D. Ha. “Modern evolution strategies for creativity: Fitting concrete images and abstract concepts.” EvoMUSART 2022.
> > > > >
> > > > > [4] B. Tjanaka, M. C. Fontaine, D. H. Lee, A. Kalkar, and S. Nikolaidis. “Training Diverse High-Dimensional Controllers by Scaling Covariance Matrix Adaptation MAP-Annealing.” RA-L 2023.
> > > > >
> > > > > [5] D. Wierstra, T. Schaul, T. Glasmachers, Y. Sun, J. Peters, and J. Schmidhuber. “Natural Evolution Strategies.” JMLR 2014.

---

### Official Review · Reviewer_83tg · 2025-10-30

**Soundness:** 3
**Presentation:** 3
**Contribution:** 2
**Rating:** 4
**Confidence:** 3

**Summary:**

This work deals with high-dimensional quality-diversity optimisation by proposing a so-called discount model search. It aims to guide exploration
with a model that provides a smooth, continuous representation of discount scalar values.

**Strengths:**

The work is well-motivated - indeed, a major issue of QD is its difficulty in solving high-dimensional problems.

The paper is well presented, and well-structured.

The result presents the effectiveness of the proposed method.

**Weaknesses:**

It seems to me that a major issue is that the method needs additional information (measures) to distinguish between solutions plateaus in QD. It would be more interesting and useful if the work could directly work on algorithm design of QD itself to address the high-dimension issue, rather than resorting to new information (which apparently can tell solutions apart). I am not sure my idea is doable, but it would be very useful if it could.

**Questions:**

I would be grateful and may potentially increase my score if the author(s) could address my comments in the weakness field.

---

> ### Author Response · Authors · 2025-11-23
>
> > It seems to me that a major issue is that the method needs additional information (measures) to distinguish between solutions plateaus in QD. It would be more interesting and useful if the work could directly work on algorithm design of QD itself to address the high-dimension issue, rather than resorting to new information (which apparently can tell solutions apart). I am not sure my idea is doable, but it would be very useful if it could.
>
> Thank you for taking the time to review our work. We appreciate your acknowledgement that our work is “well-motivated” and that our paper is “well presented.” Regarding the comments you have raised in the weakness field, we wish to clarify that our work indeed focuses on algorithm design and does not introduce additional measures or information.
>
> To elaborate, our work assumes that a given QD problem has a high-dimensional measure space. For instance, this could arise if a user has multiple image features over which they seek diversity. Another motivating use case is the QDDM setting, where the measures are high-dimensional data like images. We note that measures are a core part of the QD problem definition, and exist in every QD problem.
>
> Given this problem of QD with a high-dimensional measure space, we design an algorithm (discount model search — DMS) that addresses distortion, which appears to be a key challenge of high-dimensional measure spaces (Section 4). Importantly, we never modify the QD problem itself; DMS receives the exact same information that any other QD algorithm would. The difference is that it uses a new algorithmic component (the discount model) to enhance the search.
>
> Regarding plateaus, we mention issues with flat discount functions when we discuss CMA-MAE and how it fails (Figure 1a), but we note that these “plateaus” of the discount function are an issue with the CMA-MAE algorithm and how it performs search, rather than being a characteristic of the problem. DMS introduces a discount model to avoid these flat discount functions and their concomitant issues.
>
> _We also wish to add that we are unclear on what is meant by using “additional information (measures) to distinguish between solutions plateaus in QD.” Could the reviewer please clarify what is meant by “additional information” and “solution plateaus”? We have interpreted these to the best of our ability, but we may have misunderstood the question._

---

### Official Review · Reviewer_uYMX · 2025-11-01

**Soundness:** 4
**Presentation:** 4
**Contribution:** 4
**Rating:** 8
**Confidence:** 4

**Summary:**

The paper addresses the challenge of "distortion" in Quality Diversity (QD) optimization, where high-dimensional measure spaces cause many solutions to map to the same regions, leading to stagnation in state-of-the-art algorithms like CMA-MAE that rely on discrete archive histograms for guiding search. The authors propose Discount Model Search (DMS), which replaces the discrete discount histogram with a continuous, learned "discount model" (a neural network) to provide smoother, more accurate improvement signals in high-dimensional spaces. Additionally, the paper introduces a new QD setting called Quality Diversity with Datasets of Measures (QDDM), allowing users to specify desired diversity via datasets (e.g., images) rather than hand-crafted functions. Experiments on standard high-dimensional benchmarks and new QDDM domains (rendering triangles to match MNIST, latent space illumination with StyleGAN3) show DMS significantly outperforms current black-box QD algorithms in terms of QD score and coverage.

**Strengths:**

- Originality & Significance in addressing high-dimensional measures: The core insight—replacing the discrete, low-resolution histogram of CMA-MAE with a continuous learned model—is a novel and effective solution to the well-documented problem of distortion in high-dimensional QD. This allows QD specifically to scale to measure spaces previously considered intractable for standard MAP-Elites-based approaches.

- Introduction of QDDM: The proposal of Quality Diversity with Datasets of Measures (QDDM) is a highly practical and significant contribution. By leveraging the "manifold hypothesis" and using datasets to define centroids for CVT archives, it opens QD to a vast array of new applications where hand-crafting low-dimensional measure functions is difficult, but data is abundant (e.g., using raw images as measures).

- Strong Empirical Results: DMS demonstrates superior performance across specifically designed high-distortion benchmarks (e.g., 10D Linear Projection) compared to strong baselines like CMA-MAE and DDS. The inclusion of complex, creative domains like LSI (Hiker) demonstrates practical utility beyond simple synthetic benchmarks.

- Clarity of Exposition: The paper is exceptionally clear. Figure 1 effectively illustrates the failure mode of discrete histograms in distorted spaces. The explanation of distortion and how it is amplified by dimensionality is well-reasoned.

- Thorough Ablations: The appendices contain valuable ablations, specifically demonstrating the critical necessity of "empty points" in training the discount model to prevent it from hallucinating explored regions.

**Weaknesses:**

- Computational Overhead: As noted by the authors, training the neural network discount model introduces computational overhead compared to the simple histogram updates of CMA-MAE. While likely acceptable given the performance gains in high dimensions, a more explicit quantification of this wall-clock time trade-off in the main text would be beneficial.

- Potential Precision Limitations: The results in the TA (MNIST) domain show DMS performing slightly (though not significantly) worse than CMA-MAE in QD Score, despite similar coverage. The authors speculate this is due to noise in the discount model interfering with fine-grained objective optimization. This suggests a potential limitation of DMS in domains where extremely precise objective optimization in already-explored regions is paramount.

- Dependence on "Empty Points": The method heavily relies on explicitly sampling "empty points" to clamp the discount model. While effective in the tested domains, one might question how this scales if the measure space is so vast that almost all points are empty. Is a fixed number $n\_{empty}$ sufficient, or does it need to scale with the measure space volume?

**Questions:**

1. In the QDDM setting, how does the approach scale with the size of the user-provided dataset? If a user provides a dataset of 1 million images, does constructing the CVT archive and finding nearest centroids become a bottleneck, and does DMS still effectively navigate such a large, sparsely populated space?

2. Regarding the "noise" limitation in TA (MNIST): Have you considered a hybrid approach where the discrete histogram is used for highly local, fine-grained optimization (if cell capacity allows), while the discount model guides broad exploration?

3. Could you provide more concrete details on the wall-clock time difference between DMS and CMA-MAE on the 10D benchmarks to better illustrate the computational trade-off mentioned in the conclusion?

---

> ### Author Response · Authors · 2025-11-23
> **Author Response (Part 1)**
>
> Thank you for your comments on our paper. We are excited that you believe DMS is “a novel and effective solution” and that QDDM is “a highly practical and significant contribution.” Below we provide our responses to the weaknesses and questions raised in your review. We have grouped together weaknesses and questions that cover similar topics.
>
> > Dependence on "Empty Points": The method heavily relies on explicitly sampling "empty points" to clamp the discount model. While effective in the tested domains, one might question how this scales if the measure space is so vast that almost all points are empty. Is a fixed number sufficient, or does it need to scale with the measure space volume?
>
> We believe that a fairly small, fixed number of empty points is sufficient for DMS. In Appendix D.2, we performed an ablation on the number of empty points, setting $n_empty = 0, 10, 100, 1000$. We found that even with $n_empty = 10$, DMS achieved much higher performance than with $n_empty = 0$. Beyond that, increasing $n_empty$ further does not seem to result in higher performance. This finding was consistent across all the LP domains, both with 2D and 10D measure spaces. Hence, we believe that the number of empty points does not matter much — even a very small number seems to be capable of “clamping down” the discount model, and adding further points is redundant. Given that these results were consistent across both 2D and 10D measure spaces, we would expect to see similar behavior in higher-dimensional measure spaces, although in very high dimensions, it may be safer to use $n_empty = 100$ instead of only $n_empty = 10$.
>
> > 1. In the QDDM setting, how does the approach scale with the size of the user-provided dataset? If a user provides a dataset of 1 million images, does constructing the CVT archive and finding nearest centroids become a bottleneck, and does DMS still effectively navigate such a large, sparsely populated space?
>
> Generally, we believe it is feasible to scale DMS to large user-provided datasets. In such cases, constructing the CVT archive would not be a bottleneck, as doing so only involves providing the centroids, which are the 1 million images. On the other hand, computing the nearest centroids for new solutions may be a bottleneck. With low-dimensional measures, it would be feasible to use efficient search structures like k-D trees. However, in high dimensions, k-D trees degrade to the same performance as complete search (brute force). Hence, we can try to use complete search to find nearest centroids, perhaps accelerated on a GPU. If this approach turns out to be too slow, we can turn to approximate nearest neighbor algorithms, like those implemented in FAISS (https://github.com/facebookresearch/faiss).
>
> Regarding the effectiveness of DMS, we believe providing more centroids amounts to modifying the resolution of the archive. Since DMS uses a smooth discount function (approximated by the discount model), we expect that changes in archive resolution should not affect the smooth representation — changes in resolution would likely be noticeable in a histogram discount function like in CMA-MAE, where higher resolution might improve performance, since the cells of the histogram would be smaller and thus less affected by distortion (i.e., the histogram would look more like a smooth function).
>
> > Potential Precision Limitations: The results in the TA (MNIST) domain show DMS performing slightly (though not significantly) worse than CMA-MAE in QD Score, despite similar coverage. The authors speculate this is due to noise in the discount model interfering with fine-grained objective optimization. This suggests a potential limitation of DMS in domains where extremely precise objective optimization in already-explored regions is paramount.
> > 2. Regarding the "noise" limitation in TA (MNIST): Have you considered a hybrid approach where the discrete histogram is used for highly local, fine-grained optimization (if cell capacity allows), while the discount model guides broad exploration?
>
> Thank you for your suggestion. A hybrid approach certainly seems feasible. For instance, we could maintain the discrete histogram and the discount model, and average the values of the two so that the overall discount function was still smooth. In general, while our current work focuses on MLPs as the discount model, an exciting direction for future work would involve exploring different discount models such as this hybrid model.

---

> > ### Author Response · Authors · 2025-11-23
> > **Author Response (Part 2)**
> >
> > > Computational Overhead: As noted by the authors, training the neural network discount model introduces computational overhead compared to the simple histogram updates of CMA-MAE. While likely acceptable given the performance gains in high dimensions, a more explicit quantification of this wall-clock time trade-off in the main text would be beneficial.
> > > 3. Could you provide more concrete details on the wall-clock time difference between DMS and CMA-MAE on the 10D benchmarks to better illustrate the computational trade-off mentioned in the conclusion?
> >
> > Below we provide the wallclock time from running all algorithms in all domains. We see that in the LP domains, the discount model training increases the wallclock time compared to CMA-MAE and other algorithms. However, in the QDDM domains, the difference with CMA-MAE is minimal, as the evaluation of solutions is the bottleneck rather than the algorithm itself. We also note that MAP-Elites and MAP-Elites (line) are the fastest algorithms because they involve the fastest operations. Namely, while CMA-MAE and DMS use CMA-ES emitters, which require manipulating a multivariate Gaussian distribution, MAP-Elites and MAP-Elites (line) involve mutations that are based on fixed Gaussian noise.
> >
> > |                   | 2D LP (Sphere)   | 10D LP (Sphere)   | 2D LP (Rastrigin) | 10D LP (Rastrigin) | 2D LP (Flat)     | 10D LP (Flat)     | Arm Repertoire   | TA (MNIST)        | TA (F-MNIST)      | LSI (Hiker)         |
> > | :---------------- | :--------------- | :---------------- | :---------------- | :----------------- | :--------------- | :---------------- | :--------------- | :---------------- | :---------------- | :------------------ |
> > | DMS               | 397.83 ± 0.37    | 876.33 ± 44.22    | 707.20 ± 21.31    | 711.08 ± 30.76     | 683.53 ± 27.73   | 723.34 ± 32.31    | 687.85 ± 27.18   | 489.90 ± 12.55    | 546.60 ± 21.46    | 4,740.24 ± 482.74   |
> > | CMA-MAE           | 121.13 ± 0.10    | 333.92 ± 1.70     | 182.07 ± 0.36     | 465.80 ± 3.81      | 165.33 ± 0.25    | 800.12 ± 0.92     | 130.83 ± 0.26    | 495.95 ± 13.12    | 525.42 ± 19.22    | 4,690.12 ± 654.47   |
> > | DDS               | 1,918.68 ± 22.42 | 3,535.78 ± 1.53   | 1,957.29 ± 0.87   | 3,598.86 ± 2.56    | 1,937.41 ± 1.51  | 3,581.79 ± 2.05   | 1,577.54 ± 1.70  | --                | --                | --                  |
> > | MAP-Elites (line) | **31.10 ± 0.10** | 231.50 ± 0.73     | **39.74 ± 0.14**  | **274.86 ± 0.62**  | **30.84 ± 0.09** | **339.73 ± 0.42** | 39.68 ± 0.10     | **247.21 ± 4.07** | **231.91 ± 0.38** | 3,470.08 ± 12.88    |
> > | MAP-Elites        | 39.01 ± 0.21     | **190.25 ± 2.46** | 51.27 ± 0.17      | 367.56 ± 1.57      | 38.41 ± 0.14     | 451.46 ± 0.41     | **32.80 ± 0.08** | 291.98 ± 1.46     | 309.01 ± 3.92     | **3,460.71 ± 5.86** |
> >
> > In our main paper, we have added Table 2 and Section 7.3 to display and analyze these results.

---

### Official Review · Reviewer_Rx9F · 2025-11-01

**Soundness:** 3
**Presentation:** 3
**Contribution:** 3
**Rating:** 6
**Confidence:** 4

**Summary:**

This paper introduces Discount Model Search (DMS), a new algorithm for quality diversity (QD) optimization that addresses the challenges of exploring high-dimensional measure spaces. Traditional algorithms like CMA-MAE rely on discrete histograms to represent discount values in measure space, which causes exploration to stagnate when multiple solutions map to the same histogram cell. DMS replaces this histogram with a continuous, learned discount model—typically a neural network, that can assign smoother and more fine-grained discount values, even for similar measures.
The paper also introduces Quality Diversity with Datasets of Measures (QDDM), where the measure space is defined by a dataset (e.g., images), enabling new applications in domains like generative modeling. Extensive experiments across benchmark tasks and new QDDM domains (e.g., Triangle Arrangement and Latent Space Illumination) show that DMS outperforms state-of-the-art QD algorithms such as CMA-MAE and DDS in both QD Score and coverage.

**Strengths:**

- The paper tackles a well-motivated problem of QD algorithms in high-dimensional measure spaces.
- The paper is well written.
- there is extensive experimentation with diverse domains and baselines.

**Weaknesses:**

- If discount values of two data points are the same, then the DMS would treat them the same way. As with every quantitative measure, by Goodhart's law, pathologies may appear when we optimize for quantitative measures. Did the authors observe any of such phenomenon? How might the authors foresee mitigating such a limitation in future?

**Questions:**

- In Figure 2, a lot of the characteristics of the person (e.g., skin tone, facial features) remain the same even though the input image looks different. And although the image looks more similar sometimes with lakes in different images, the person looks more different. Do the authors have any insight or analysis on why this happens?
- How do the authors foresee the algorithm to be applicable to text domains? Could we reuse pretrained LLMs as the DMS?
- Is there a possibility of transferring the trained DM across tasks?
- Could the same mechanism be applied to the fitness function?

---

> ### Author Response · Authors · 2025-11-23
> **Author Response (Part 1)**
>
> Thank you for taking the time to review our paper. We appreciate your comments that our paper is “well-motivated” and has “extensive experimentation.” Below we provide our responses to the weaknesses and questions mentioned in your review.
>
> > If discount values of two data points are the same, then the DMS would treat them the same way. As with every quantitative measure, by Goodhart's law, pathologies may appear when we optimize for quantitative measures. Did the authors observe any of such phenomenon? How might the authors foresee mitigating such a limitation in future?
>
> We do not believe it is problematic for DMS to encounter points with the same discount values, because DMS does not necessarily treat such points the same way. To elaborate, DMS ranks points by archive improvement (Algorithm 1, line 11 and line 17), which is the difference between the objective value of the solution and the discount value output by the discount model. To obtain identical archive improvements and thus make DMS treat the solutions more similarly, the solutions would need to have identical objective and discount values. Even then, having the same archive improvement is only problematic if the solutions are close together in measure space, because then it would be unclear which measure space direction DMS should pursue to improve the archive (similar to the scenario shown in Figure 1a). Overall, we believe that the scenario of having two completely indistinguishable solutions is unlikely with DMS.
>
> We wish to clarify that discount values are an internal component of DMS that help guide the search. Discount values and archive improvement are not external quantitative measures, and they are not included in the QD problem definition. On the other hand, prior work [1] has shown that optimizing for archive improvement directly corresponds to optimizing for the QD objective. As with any other objective, it is certainly possible to encounter pathologies such as local minima. One step to addressing such limitations could be to try alternative emitters — DMS uses CMA-ES-based emitters, but emitters based on variants of CMA-ES [2] or Natural Evolution Strategies [3] are certainly feasible.
>
> > In Figure 2, a lot of the characteristics of the person (e.g., skin tone, facial features) remain the same even though the input image looks different. And although the image looks more similar sometimes with lakes in different images, the person looks more different. Do the authors have any insight or analysis on why this happens?
>
> We suspect that this phenomenon occurs because of our usage of the CLIP score for associating hiker images with landscape images. The CLIP score measures the semantic similarity of two images, so similar landscapes may be associated with vastly different hikers, and similar hikers may be associated with vastly different landscapes. Furthermore, we speculate that the CLIP score is imperfect, and slight changes to the appearance of a hiker or landscape may result in sharp changes to the semantic meaning.
>
> > How do the authors foresee the algorithm to be applicable to text domains? Could we reuse pretrained LLMs as the DMS?
>
> Indeed, in text domains, we could use a pretrained LLM to embed text for our discount model. Then, we would attach an MLP head to the discount model that takes in the embeddings and outputs the scalar discount values. While running DMS, we could train that head while keeping the pretrained LLM frozen. This approach is similar to what is done in the LSI (Hiker) domain, where we embed the images with the vision transformer from CLIP and then pass that embedding to an MLP head.
>
> As a more concrete example, let us consider red-teaming of LLMs, which has recently become a popular application for QD algorithms [4]. We could frame this problem as a QDDM domain where we seek to find prompts that elicit harmful outputs from an LLM while resembling safe prompts. Our measure space would be the entire space of text, and the desired measure values would be a dataset of safe prompts. During search, we would insert into the archive by matching each harmful prompt that we find with the safe prompt that they most closely resemble (either using semantic similarity or a metric like BLEU score). The LLM-based discount model would then reflect which areas of the measure space have already been explored. The final piece of this application lies in generating the prompts. Doing so would require converting the continuous solutions output by the CMA-ES emitters into pieces of text — this may be possible with techniques such as soft prompting.

---

> > ### Author Response · Authors · 2025-11-23
> > **Author Response (Part 2)**
> >
> > > Is there a possibility of transferring the trained DM across tasks?
> >
> > The discount model cannot be transferred across tasks because its purpose is to guide search during a single run of DMS. To elaborate, the discount model provides a smooth approximation of the current discount function. On each iteration, the model is updated to reflect the new discount function, which enables it to continue guiding the emitters.
> >
> > Once DMS has concluded, the discount model essentially indicates which areas of the archive have been explored/optimized and which ones remain to be explored/optimized. If we were to transfer this discount model to a new task, DMS there would be starting with an empty archive, yet the discount model would be indicating that some areas of this archive have already been explored. In short, the discount model is tied to each run of DMS and thus cannot be transferred across runs.
> >
> > > Could the same mechanism be applied to the fitness function?
> >
> > Measures and objectives (fitness) serve quite different purposes in QD, as we seek to find solutions with every possible value of the measure, while for the objective, we only care about finding the highest objective values. The discount model is designed to address the issue of measure space distortion, which makes it hard to find solutions with every possible value of the measure function. If we applied a similar mechanism to the objective function, we would essentially be searching for solutions with every possible objective value, rather than only solutions with the highest objective values. We suspect the mechanism would work, but we believe it would be addressing a slightly different problem than QD.
> >
> > [1] M. C. Fontaine and S. Nikolaidis. “Covariance matrix adaptation map-annealing.” GECCO 2023.
> >
> > [2] B. Tjanaka, M. C. Fontaine, D. H. Lee, A. Kalkar, and S. Nikolaidis. “Training Diverse High-Dimensional Controllers by Scaling Covariance Matrix Adaptation MAP-Annealing.” RA-L 2023.
> >
> > [3] D. Wierstra, T. Schaul, T. Glasmachers, Y. Sun, J. Peters, and J. Schmidhuber. “Natural Evolution Strategies.” JMLR 2014.
> >
> > [4] M. Samvelyan, S. C. Raparthy, A. Lupu, E. Hambro, A. H. Markosyan, M. Bhatt, Y. Mao,
> > M. Jiang, J. Parker-Holder, J. Foerster, T. Rocktäschel, and R. Raileanu, “Rainbow teaming: open-ended generation of diverse adversarial prompts,” NeurIPS 2024.

---

### Meta-Review · Area_Chair_3Zmj · 2025-12-27

**Summary:**

In this work the authors propose a continuous (MLP-based) measure model for quality diversity optimization tasks, DMS. This overcomes limitation with the state of the art methods such as CMA-MAE that discretize the measure space, and suffer from lower diversity solutions in higher-dimensional measure spaces. They also introduce an alternative QD formulation (QDDM) that allows users to specify diversity exemplars (as opposed to measures), which in some domains (e.g. high-dimensional image generation) may be a more natural way to impose diversity. The authors present compelling empirical validation of their methods (DMS and QDDM).

Strengths:

- DMS is a well founded and useful solution to the dimension-distortion problem suffered by many QD methods in higher-dimensions.
- QDDM is a practical and significant contribution for higher-dimensional settings or where formulation of measure functions is difficult/tedious.
- Strong empirical results
- Cleary written
- Thorough ablation testing


Weaknesses (mostly resolved):

- Concerns of increased computational burdens over baselines for continuous discount model training and for QDDM with millions of exemplars
- In some situations, it seems like histogram based approaches outperform DMS
- Use of MLPs as opposed to other regression heads for measure function estimation
- Only 16 dimensional measure space (in non QDDM settings) considered
- Dependence on “empty points” hyperparameter, and the relationship with it and problem dimensionality

This is a very interesting work, that presents a significant contribution. Most of the reviewer concerns were addressed in the discussion period by either clarifying discussion, additional experiments or ablations (that have been added to the paper). I believe this paper will be a great addition to ICLR.

**Reviewer Concerns:**

Concerns addressed:

- Computational overhead concerns were addressed by explicit wall clock times being reported for the experiments. DMS is slower than some baselines (especially MAP-elites) but the performance I think justifies the increased cost. Furthermore a comprehensive discussion about the number of QDDM exemplars was given.
- In some situations, it seems like histogram based approaches outperform DMS. This does not seem to be a big issue, and the authors mention this could be solved with a hybrid approach, which may be a future work direction.
- Use of MLPs as opposed to other regression heads for measure function estimation -- the authors state that the MLP is a simple model, that is relatively easy to train, and works in high dimensions; where some of the suggested alternatives may not. Furthermore, it is often operating on top of a pre-trained model.
- Only 16 dimensional measure space (in non QDDM settings) considered -- the authors added 20 and 50 dimensional experiments.
- Dependence on “empty points” hyperparameter, and the relationship with it and problem dimensionality -- an ablation was presented to address this point, recommending $n_{empty} = 10$ being a fairly robust choice.



Concerns unaddressed:

No major concerns unaddressed as far as I can tell.

**Reviewer Scores:**

Rx9F: 6 -> 8 all concerns addressed

uYMX: 8 ->  8 or even 10 as all their concerns were addressed

83tg: NA, I don’t believe this review is of sufficient quality (it does not present a constructive criticism of the work as is), so I am not considering it.

TnHM: 4 -> 6 or 8 as this reviewer's concerns were thoroughly addressed.

As such I think the final score range would be 7.33 to 8.

---

### Decision · Program_Chairs · 2026-01-26

Accept (Oral)